


Climate
of the Past

# Technical note: Optimizing the utility of combined GPR, OSL, and lidar (GOaL) to extract paleoenvironmental records and decipher shoreline evolution

**Amy J. Dougherty**[1], **Jeong-Heon Choi**[2], **Chris S. M. Turney**[3], and **Anthony Dosseto**[1]

[1]School of Earth, Atmospheric and Life Sciences, University of Wollongong, Wollongong, 2522, Australia
[2]Department of Earth and Environmental Sciences, Korea Basic Science Institute, Ochang, 28119, South Korea
[3]Palaeontology, Geobiology and Earth Archives Research Centre (PANGEA) and the Australian Research Council Centre of
Excellence for Australian Biodiversity and Heritage (CABAH), School of Biological, Earth and Environmental Sciences,
The University of New South Wales, Sydney, 2052, Australia

**Correspondence:** Amy J. Dougherty (adougher@uow.edu.au)

**Abstract.** CE1 Records of past sea levels, storms, and their impacts on coastlines are crucial for forecasting and managing future changes resulting from anthropogenic global warming. Coastal barriers that have prograded over the Holocene preserve within their accreting sands a history of storm erosion and changes in sea level. High-resolution geophysics, geochronology, and remote sensing techniques offer an optimal way to extract these records and decipher shoreline evolution. These methods include light detection and ranging (lidar) to image the lateral extent of relict shoreline dune morphology in 3-D, ground-penetrating radar (GPR) to record paleo-dune, beach, and nearshore stratigraphy, and optically stimulated luminescence (OSL) to date the deposition of sand grains along these shorelines. Utilization of these technological advances has recently become more prevalent in coastal research. The resolution and sensitivity of these methods offer unique insights on coastal environments and their relationship to past climate change. However, discrepancies in the analysis and presentation of the data can result in erroneous interpretations. When utilized correctly on prograded barriers these methods (independently or in various combinations) have produced storm records, constructed sea-level curves, quantified sediment budgets, and deciphered coastal evolution. Therefore, combining the application of GPR, OSL, and lidar (GOaL) on one prograded barrier has the potential to generate three detailed records of (1) storms, (2) sea level, and (3) sediment supply for that coastline. Obtaining all three for one barrier (a GOaL hat-trick) can pro-

vide valuable insights into how these factors influenced past and future barrier evolution. Here we argue that systematically achieving GOaL hat-tricks on some of the 300+ prograded barriers worldwide would allow us to disentangle local patterns of sediment supply from the regional effects of storms or global changes in sea level, providing for a direct comparison to climate proxy records. Fully realizing this aim requires standardization of methods to optimize results. The impetus for this initiative is to establish a framework for consistent data collection and analysis that maximizes the potential of GOaL to contribute to climate change research that can assist coastal communities in mitigating future impacts of global warming.

## 1 Introduction

Global warming is a major driver of sea-level rise and is projected to increase the frequency and magnitude of storms, but the extent of these changes and their impacts on vulnerable sandy coastlines is uncertain (IPCC, 2013). Paleoenvironmental records of sea level and storms as well as the evolution of shorelines throughout the Holocene can provide insight into future environmental and societal impacts (Little et al., 2017; Caseldine and Turney, 2010). Coastlines that have a positive sediment budget and space available to accommodate it have built seaward through time, forming strand plains comprising a series of foredune–beach ridges

(Fig. 1a). These accreted coastal sands preserve a history of sea-level change, storm impacts, and sediment supply within their stratigraphy. The resulting coastal systems are called prograded barriers, and they have been studied for over a half-century to decipher their evolution and extract paleoenvironmental records (e.g. Bernard et al., 1962; Curray et al., 1969; Schofield, 1985; Thom et al., 1981). Over the past few decades, more traditional methods have been augmented by state-of-the-art remote sensing, geophysical, and geochronological techniques (e.g. Dougherty et al., 2016; Tamura, 2012). For instance, two-dimensional topographic surveys of dune ridges (Fig. 1a) were expanded laterally by 3-D digital terrain models produced using light detection and ranging (lidar) (e.g. Gutierrez et al., 2001). Generalized stratigraphic cross sections interpolated between cores (Fig. 1a) have been filled in with detailed dune, beach, and nearshore structures from high-resolution ground-penetrating radar (GPR) (e.g. van Heteren et al., 1998). Finally, optically stimulated luminescence (OSL) directly dates when beach and dune sand was deposited (e.g. Jacobs, 2008), eliminating the extrapolation of radiocarbon ages using isochrons (Fig. 1a). The utility of combining GPR, OSL, and lidar on prograded barriers has been demonstrated successfully in previous studies (e.g. Clemmensen et al., 2014; Mallinson et al., 2008; Muru et al., 2018; Nooren et al., 2017; Timmons et al., 2010; Tõnisson et al., 2018). Foreseeing the future use and potential of these combined methods, this technical note outlines a systematic and semi-standardized structure for data collection and interpretation. The strategy is that with a large enough dataset of similarly studied prograded barriers around the world, local to global forcing on coastal evolution can be better deciphered.

There has been a steady uptake in the utilization of these geophysical, geochronological, and remotely sensed data since the decades when the applications were first introduced. Recently, there has been notable proliferation in their use associated with the ease with which this data are able to be acquired (as lidar becomes more available, GPR more user-friendly, and OSL more accessible). However, as Christopher Hein (personal communication, 19 March 2018) succinctly highlighted, some tools like GPR or pre-processed lidar data are perhaps easy to use, but not easy to use well. These techniques are all specialty fields of science in their own right and collaboration between experts in these different disciplines can avoid common pitfalls. This is important not just to ensure that the data are as precise and accurate as possible, but also that the results (or lack thereof) are presented in such a way that they do not mislead interpretations. This is not always straightforward with these types of high-resolution datasets as the detail and volume can mask or overwhelm significant aspects–features; analogous to obscuring both the forest (barrier evolution) and the trees (individual beachfaces). Therefore, it is important to be intentional with the questions being addressed using a dataset and diligent

about the interpretations as well as implications drawn from it.

Studies have shown that utilizing these approaches on prograded barriers, independently or in various combinations, can (1) decipher frequency–intensity storm records (e.g. Buynevich et al., 2007; Dougherty, 2014; Nott and Hayne, 2001), (2) construct sea-level curves (e.g. Nielsen et al., 2017; Rodriguez and Meyer, 2006; van Heteren et al., 2000), (3) quantify sediment budgets (e.g. van Heteren et al., 1996; Bristow and Pucillo, 2006; Choi et al., 2014; Dougherty et al., 2015), and (4) decipher coastal evolution (e.g. Barboza et al., 2009; Costas and FitzGerald, 2011; Hein et al., 2016). Combining GPR, OSL, and lidar (GOaL) on certain systems offers the possibility to determine a history of storms, sea level, sediment supply, and their impact on shoreline evolution all at once. Given the increased prevalence of these techniques and the existence of 300+ prograded barriers located around the world (Scheffers et al., 2012), a systematic application of GOaL to decipher coastal evolution can also detect local patterns of sediment supply, regional records of storms, or global changes in sea level. The larger-scale records have the potential to be used like and combined with other climate proxy records. The possibilities necessitate standardizing important parts of this methodological approach to optimize results. The aim of this article is threefold: (1) present a basic introduction to the capabilities of GOaL individually, (2) provide a simple strategy that logically utilizes information from each technique to optimize the resulting GOaL dataset, and (3) highlight the possibilities and pitfalls associated with the data to maximize the combination of GOaL on prograded systems.

## 2   GOaL methodological approach

With each GOaL technique producing such high-resolution data, how they are collected and presented can affect the results or interpretations. This section explains a simple methodological approach to maximize the volume and detail of GOaL from prograded barriers. These methods are introduced in the order that they are recommended to be utilized, with a brief statement on the logic of applying each technique in the three-step methodology. Specifics on the different techniques, instrumentation, or settings and parameters are not discussed. There is already a large body of literature about these different methods and their utilization in the coastal settings referenced within each section. The type of equipment or method used is usually reliant on what is available to the researcher and ideal settings are site specific. Furthermore, coastal researchers often rely on other experts in the fields of remote sensing, geophysics, and geochronology to collect the data or even utilize previously published results. This technical note is not a "how-to" guide with specifics for acquiring and analysing each dataset. Rather, it discusses the

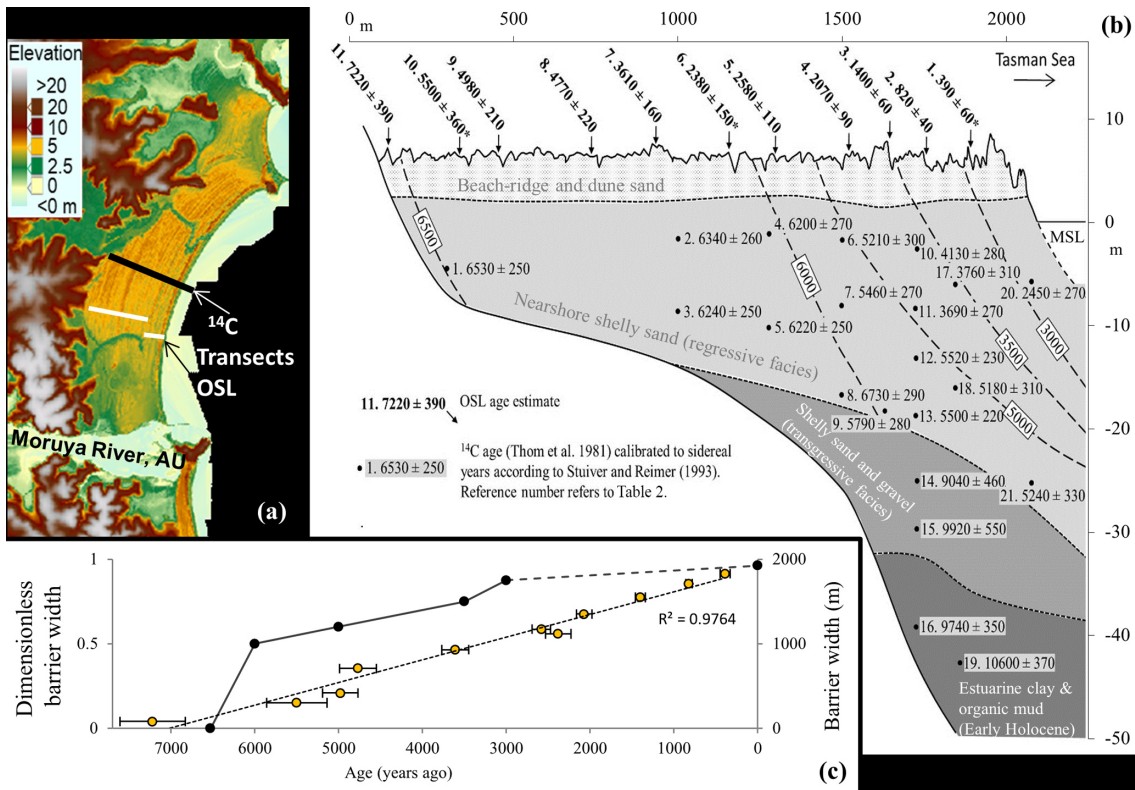

**Figure 1. (a)** Lidar data on the prograded barrier system near Moruya, Australia, with the location of the transects where [14]C and OSL samples were collected. **(b)** Stratigraphic cross section of Moruya Barrier displaying radiocarbon (cal yr BP) and OSL (years before 2012–2013) chronologies (Oliver et al., 2015; Thom et al., 1981). **(c)** Diagram of barrier width as a function of OSL (yellow dots with black circles) and radiocarbon (black dots) ages. The new OSL chronology shows that progradation has been much more linear than was previously thought using radiocarbon. Figure modified from Dougherty et al. (2016) and Oliver et al. (2015).

potential of combining these techniques and offers a practical approach to optimize the dataset.

This article advocates that any of these high-resolution datasets, when collected and analysed correctly, improves our understanding of coastal evolution. However, these higher-tech approaches do not negate the use of more traditional techniques, like using radiocarbon dating where suitable material exists as it is faster and cheaper to process. There are many examples in which research on chenier plains or coarse clastic beach ridges has used radiocarbon, OSL, GPR, and lidar in various combinations to determine their evolution and a history of storms or sea level (e.g. Billy et al., 2015; Dougherty and Dickson, 2012; Hein et al., 2016; Hijma et al., 2017; Long et al., 2012; Morton et al., 2000; Neal et al., 2002; Weill et al., 2012). While the GOaL approach proposed in this paper is geared toward the more prevalent sandy prograded barriers, it could easily be applied to (and compared with data from) these other types of coastal settings. In any environment, the utilization of remote sensing techniques necessitates, rather than negates, the use of established methods (e.g. coring, augering, outcrop mapping, and/or topographic profiling) to ground-truth the data.

Ultimately, these means of ground-truthing remotely sensed data are an integral component of (and should be embedded in) GPR and lidar methodologies, whether they are used independently or as part of the GOaL approach. The GOaL methodology may not be possible or ideal for all sites; however, when these techniques are able to be used, this article aims to provide insight on how to optimize their utility to extract paleoenvironmental records and decipher impacts of storms, sea level, and sediment supply versus accommodation space. Results from published studies are used to demonstrate the capabilities of GOaL independently, as well as the advantages of combing them in the suggested order.

## 2.1 Lidar

Documenting barrier morphology and coastal setting is a vital first step to understanding shoreline evolution. Airborne lidar uses scanning laser altimetry as a survey method of obtaining topographic information for coastal dunes and intertidal areas above the low-water mark (Fig. 2). Aircraft-mounted sensors combine global positioning systems (GPSs) and laser range finders to remotely map the surface of the Earth over areas tens to hundreds of kilometres in extent,

with a horizontal resolution of 1 m or less and a vertical accuracy of 0.10 to 0.15 m. Detailed information about the elevation of the land surface and vegetation is acquired by emitting laser pulses, which reflect off objects and produce a backscatter recorded by the sensor. In addition to a "travel time" for each pulse and subsequent return signal, an intensity of reflectance is also often measured and used to identify vegetation canopy versus ground surfaces. Drones equipped with lidar are being explored as a lower-cost option to acquire coastal data, but it is still expensive and requires experience to use (including a pilot license in some airspace) (Klemas, 2015). This section does not discuss the complex details of how to collect or process lidar, but rather optimally utilizing professionally acquired and processed data.

Traditionally, air photographs, satellite images, and topographic profiles have been used to assess coastal systems as well as plan fieldwork. The advent of platforms like Google Maps, Google Earth, NASA Worldview, and NASA Word Wind, which provide free imagery collected over time, bolstered the amount of data available (Fig. 2). Lidar penetrates the vegetation that often obscured details of the morphology in aerial imagery and removes this obstruction during processing. Digital terrain models created form lidar data refine the morphology, detecting subtle dune topography. This dataset can be used to extract topographic profiles and calculate the volume of barrier sediment supplied above mean sea level (Dougherty et al., 2015, 2012; Oliver et al., 2014). The classic prograded barrier system located near Moruya, Australia, offers an example of the detail and lateral extent mapped in lidar (Fig. 1a) compared to the original two-dimensional topographic profile (Fig. 1b). The lidar captures the uniform shoreline progradation represented by the series of beach–foredune ridges (yellow with high crest in red, Fig. 1a) as well as interactions from inlets, tidal creeks, and open ocean (green and off-white colours, Fig. 1a). Detailed barrier morphology derived from lidar can be used to (1) target areas modified by natural and human processes to understand their impact or (2) avoid them to isolate the influence of storms, sea level, and sediment supply versus accommodation space.

The display or rendering chosen to analyse and present lidar data can impact interpretations. Since coastal systems are relatively low-lying features, the elevation scale range and colour scheme chosen should at least define the barrier from intertidal areas (done using cool and warm colours in Fig. 1a). In more complicated systems the display should be such that important changes in the surrounding geologic setting or within the dune morphology are easily discernible (Fig. 2). Once the lidar is optimally rendered, these remotely sensed data needs to be ground-truthed to detect any errors in data acquisition or processing deficiencies (Gutierrez et al., 2001). This can be done in the field by checking the elevations using traditional survey equipment such as levels and total stations or real-time kinematic (RTK) GPS. It is acknowledged that lidar is not available for large parts of the

world and other technologies for mapping morphology exist that may be easier or less expensive (e.g. drone-based "structure from motion"; Christopher Hein, personal communication, 14 December 2019). When using another comparable method to acquire high-resolution, large-spatial-extent topographic data, it is equally important to ground-truth and render them properly.

Augmenting air photos or satellite images with lidar provides a more complete understanding of the geologic setting to contextualize and understand coastal evolution as well as plan fieldwork. A Google Earth image of the Rangitaiki Plains in New Zealand displays a filled coastal embayment that has a prominent series of foredune ridges behind the present-day shoreline (Fig. 2). The lidar data in Begg and Mouslopoulou (2010) show that the infilling did not occur by uniform shoreline progradation, like at Moruya (Fig. 1), but is a rather complex evolution influenced by tectonic and riverine processes. This lidar imaged the modern prograded barrier island that formed after the area experienced $\sim 5$ m of subsidence between 2.1 and 1.72 kyr ago (Begg and Mouslopoulou, 2010: circled in black in Fig. 2d). The lidar data also identified remnants of prograded foredune ridge sequences preserved in the eastern section of the embayment (white circles in Fig. 2d). The detail revealed that the easily erodible beach and dune sands along the seaward side of these prograded sequences appear to have been modified. However, their landward extent does not appear eroded, especially the oldest two sequences that display the same natural transition to back-barrier deposits identified in the modern barrier island (documented by cores in Pullar and Selby, 1971). To test the hypothesis that these features formed similarly to the modern analogue, resulting in a unique set of prograded barrier islands, the lidar data were used to determine the best location to collect GPR transects (grey lines in Fig. 2d). Toggling between overlain lidar and Google Earth images provided pre-field reconnaissance of obstacles (trees, houses, etc.) to consider logistics when targeting each specific profile. Given how rapidly and drastically coastal landscape changes, selecting the Google Earth image dated closest to when the lidar was collected is instrumental to providing good correlation in the overlay. It is also optimal to publish the lidar data augmented with aerial imagery when possible. This is useful for the reader to analyse barrier morphology in relation to shallow subaerial offshore, inlet, estuary sediment deposits, and/or human modification that is sometimes not captured in the lidar.

## 2.2  GPR

Once the surface morphology is analysed, the next step to determine how a barrier formed is to study the history preserved in the shallow subsurface. The lidar data should be used to make informed decisions on where best to acquire detailed stratigraphy using geophysics. Ground-penetrating radar (GPR) is a high-resolution geophysical technique can

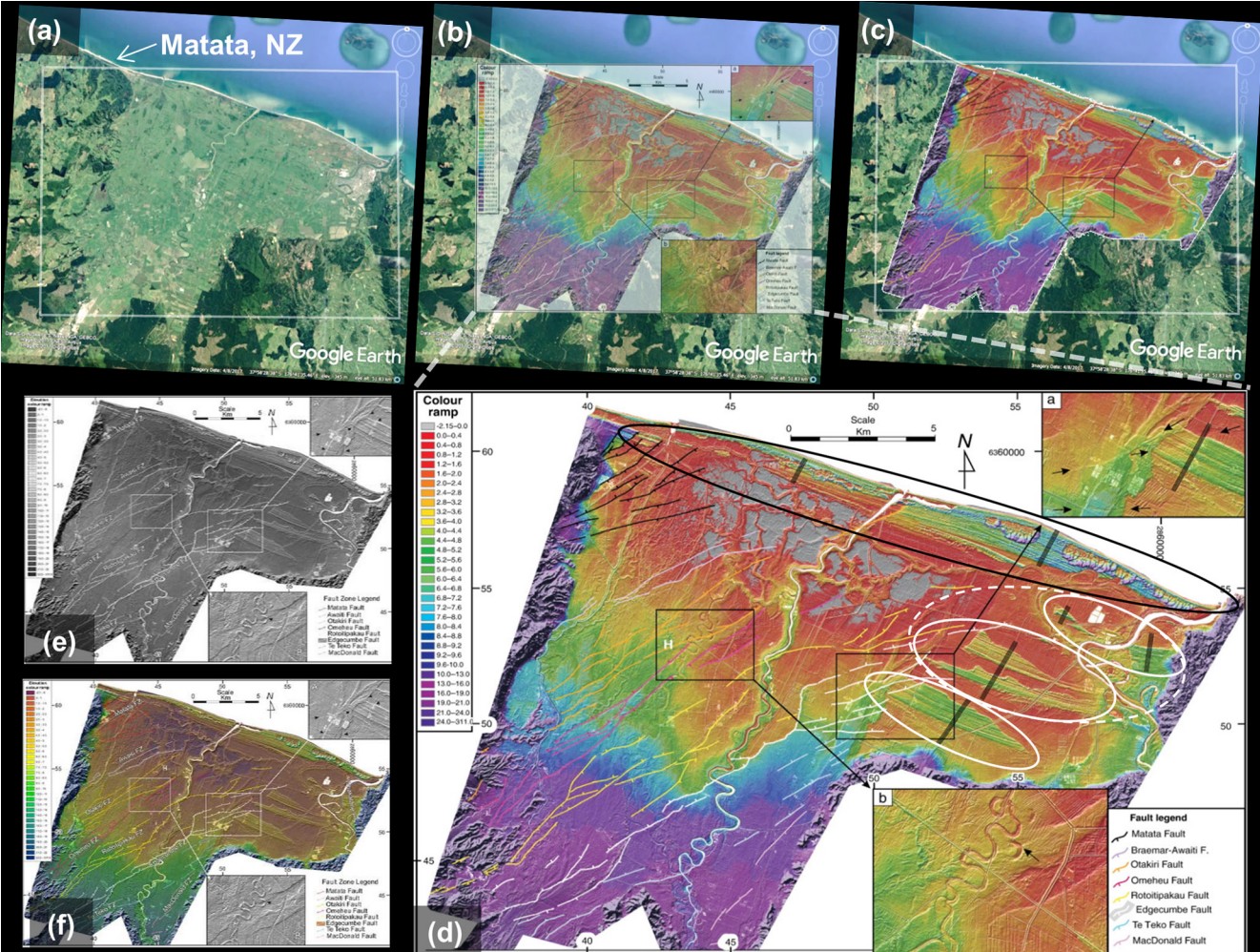

**Figure 2.** **(a–c)** A Google Earth image of Rangitaiki Plains, New Zealand, overlain with lidar shows complex infilling of this coastal embayment. **(d)** The modern coastline displays a prograded barrier island (black oval). Faulting and river dynamics appear to have eroded the central and western portion of older prograded barrier islands preserved in the eastern portion of the embayment (white ovals). Note the difference in the rendering of the lidar data and how the colour scheme chosen can either highlight the barrier structures **(b–c)** or blend them with the background **(e–f)**. Lidar modified from Begg and Mouslopoulou (2010).

image dune, beach, and nearshore facies with decimetre resolution over kilometres of coast (e.g. Buynevich et al., 2009; Barboza et al., 2011). To achieve subsurface imaging, GPR emits short pulses of electromagnetic energy (microwave radiation) into the ground (Jol et al., 1996). These transmitted high-frequency radio waves are sensitive to the electrical conduction properties of the material being penetrated (dielectric permittivity) and differences in permittivities cause them to be reflected, refracted, or scattered back to the surface. A receiving antenna records variations in the return signal, detecting changes in material properties of subsurface structures and facies by travel time within the waveform. Individual waveforms display changes within the subsurface by recording a wave-amplitude spike at a stratigraphic boundary surface. Collecting GPR along a transect line stacks individual wave traces laterally such that low wave-amplitude signals represent homogenous sediments, and increase in amplitude is associated with greater contrast in sediment characteristics (e.g. change in water content, mineralogy, grain size, sorting, etc.). The variation in waveform detects changes that occur at stratigraphic boundaries, as peaks of high amplitude merge to form strong reflection surfaces. It also detects more subtle changes within the facies, with lower-amplitude peaks forming medium to weak reflections (Fig. 3).

Of the three GOaL techniques, GPR is the most easily accessible and affordable method for coastal geologists to collect and process data independently. The ability to buy or rent a GPR increased as their operation became more user-friendly (e.g. from completely analogue systems with a stylus recorder to partially digital systems using DOS on a control unit or laptop computer, and now some are complete with digital antennas using Bluetooth communication run through

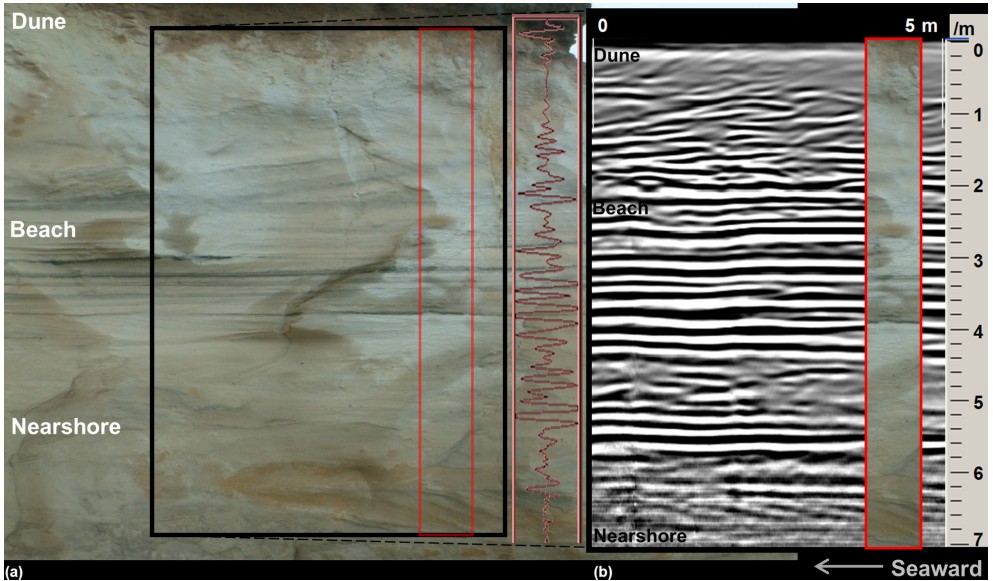

**Figure 3. (a)** Photograph of a scarp that cross-cuts a prograded Pleistocene barrier located near One Tree Point, New Zealand. This outcrop displays the small-scale stratigraphy of the barrier facies: dune, beach, and nearshore. **(b)** Transect of GPR data collected along the top of this outcrop that accurately maps the sedimentary beds exposed and records the internal barrier structure in detail. A single waveform is displayed between the GPR data and the corresponding outcrop to exemplify how the wave-amplitude spikes correspond to changes in the stratigraphy and laterally form the strong or weak reflections in the geophysical data. The section of the outcrop photo (outlined in red) is overlain on the GPR data to demonstrate the need to ground-truth the geophysical data with cores to determine the cause of the reflections. Note that all of these overlays are approximate as GPR had to be collected a small distance from the cliff to minimize edge effects within the geophysical data. Figure modified from Dougherty and Nichol (2007).

simple Windows interfaces on tablets). Currently there are many brands and configurations of different ages in use as well as a variety of software packages that can be utilized to process their data. It is not within the scope of this article to discuss all the differences in components, set-up configurations, settings, processing steps, and terminology. This article advocates for neither a particular unit, antenna, and software nor specific settings or a certain set of processing steps, as multiple variations produce similar high-resolution images of barrier stratigraphy when used correctly. Ultimately the type of equipment used for a certain project likely depends on what is available to the researcher. Novice users should utilize the extensive literature that exists on GPR and its use in coastal settings (e.g. Bristow and Jol, 2003; Buynevich et al., 2009). In addition to acquiring standard knowledge of GPR and the basics of processing, it is useful to research previous publications that use the same equipment that is available to the scientist for specifics. It is also important to reiterate that when starting out it is best to collaborate or consult with someone who has experience with GPR, not just for acquisition and processing, but also especially for interpreting the data. For use in GOaL, it is expected that there is a level of competency in GPR data collection, basic processing, and interpretation.

Initial cross-sectional models of prograded barriers display generalized shallow stratigraphy with large-scale sub-surface facies boundaries interpolated from drill core data and isochrons extrapolated from $^{14}$C age samples (e.g. Bernard et al., 1962; Curray et al., 1969; Thom et al., 1978: Fig. 1b). The electromagnetic properties of sandy barriers are ideal for producing excellent GPR images because of the high resistivity of the sediment opposing the flow of electrical current (Leatherman, 1987). Collecting GPR across entire prograded barriers can extract high-resolution stratigraphic records providing a continuous cross-sectional view of barrier architecture that detects small-scale features and large-scale facies boundaries previously unrecognized in point source core data (e.g. Fitzgerald et al., 1992; Jol et al., 1996; van Heteren et al., 1998). A unique outcrop of a Pleistocene prograded barrier in One Tree Point, New Zealand, illustrates the sensitivity of GPR in detecting stratigraphy (Fig. 3). The geophysical record shows how the heavy mineral beachfaces create the strongest reflections between 2 and 5 m. Medium-strength reflections are detecting the more diffuse heavy mineral concentrations within the dune sequence (0–2 m of depth) and in the cross-bedding preserved as a bar migrated onshore in the nearshore (6–7 m of depth). The weak, reflection-free areas in the dune and nearshore represent homogenous deposits. However, GPR uncovers structure in the fine-grained, well-stored, quartz dune sand at the top that would have been otherwise invisible to the naked eye.

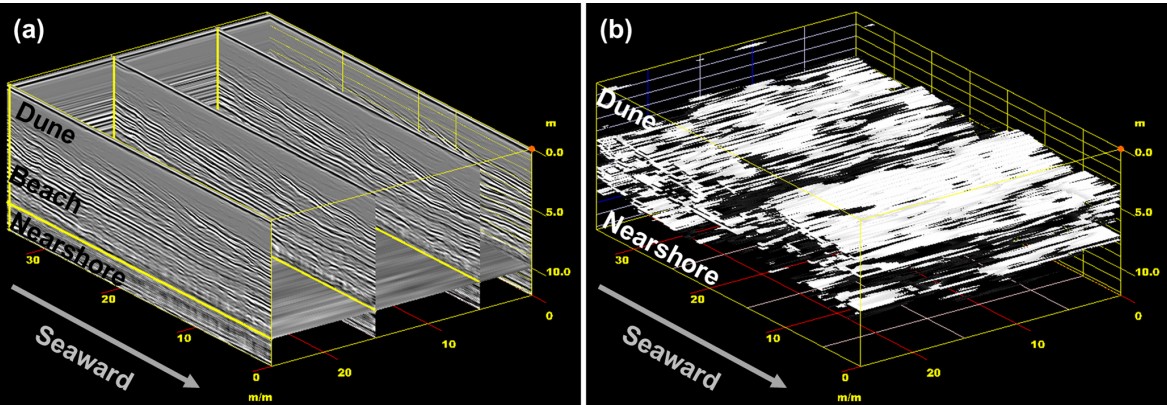

**Figure 4. (a)** Fence diagram showing some of the GPR transects collected in a grid configuration that identify barrier facies. **(b)** 3-D model of storm-eroded beachface stratigraphy constructed by isolating the most prominent reflections, shown in white, and interpolating between the transect lines. Figure modified from Dougherty (2011).

Ground-penetrating radar can detect differences such as compaction and/or water content, allowing stratigraphy to be more obvious in the geophysical records. For example, the prominent reflections between 5 and 6 m identify the transition in sands between beach and nearshore facies that is otherwise not detectible without grain-size analysis (Fig. 3). The ability of GPR to detect individual beachfaces as well as their boundaries with dune and nearshore facies enables them to be mapped throughout a prograded barrier. Mapping the beachfaces through time allows their elevation to be used as a sea-level proxy (e.g. van Heteren et al., 2000; Costas et al., 2016) and their geometry to produce storm records (e.g. Goslin and Clemmensen, 2017; Lindhorst et al., 2008). Because GPR is sensitive to subtle changes in the subsurface, the record must be ground-truthed using cores, augers, or outcrops in order to verify barrier facies and boundaries (e.g. Costas and FitzGerald, 2011; Hein et al., 2013, 2016). Additionally, topographic profiles of the present-day beach and sediment samples from each facies should be collected, preferably capturing both storm and swell geometry and sedimentology. Typically, erosion concentrates storm lag deposits on the steepened upper beachface and/or flattened lower beachface, which causes high-amplitude reflections that are more prominent than the low-amplitude signature of the homogenous berm sands that accrete during intervening swell conditions. Mapping these distinct geophysical signatures throughout the barrier enables storm records to be extracted (Buynevich et al., 2007, 2004; Dougherty et al., 2004). As a whole, the high- to medium-amplitude beachface signatures stand out compared to the weak or reflection-free signals in the dune and nearshore facies (Fig. 4). This contrast allows beachface elevation to be used as a proxy for sea level (e.g. van Heteren et al., 2000; Rodriguez and Meyer, 2006; Dougherty, 2014). While lidar can be used to apply a coarse topographic correction to the GPR data, it is recommended that precise topographic profiles be surveyed in the field and tied directly to the GPR transect, in particular if the aim is to extract sea-level and storm records.

In order to delineate barrier facies and individual beachfaces it is fundamental to ensure that the amplitude of the waveform peak relates to the contrast within the stratigraphy (e.g. the strongest reflections are the storm-eroded beachfaces and the weakest are homogeneous dune sands). The waveform amplitudes can be adjusted using what is referred to as a gain control. Unlike other basic processing steps, there has been relatively little discussion about gain in the literature; but the fact that incorrectly gained data can impact interpretations warrants attention. The correct application of gain is not just important to accurately represent and interpret barrier stratigraphy, but also critical to the extraction of sea-level and storm records from it. Gain represents the value by which the scaled waveform data are multiplied to get the output data. It is important to adjust the gain according to the core, auger, and/or outcrop data as low gain makes all reflections weak and high gain makes all reflections strong. This lack of contrast makes it hard to distinguish different barrier facies boundaries (used as a sea-level proxy), let alone individual beachfaces (to determine eroded paleo-beachfaces used to construct a storm record). It is also important to keep in mind that individual changes in the subsurface result in double peaks within the waveform, which are presented in the GPR record as prominent coupled lines (demonstrated in Fig. 3 as white and black or black and white, depending on normal or reverse polarity). This means that not all lines on a GPR record represent changes in the subsurface (e.g. Fig. 3). As such, it is not recommended to simply trace every line when interpreting GPR data. Three-dimensional grid modelling can be used to visualize how good gain control can distinguish barrier facies boundaries (Fig. 4a) and isolate storm-eroded beachfaces by interpolating the highest-amplitude reflections among a series of shore-perpendicular transects (Fig. 4b). The use of 3-D models is not necessary

for extracting sea-level and storm records, but could be useful in studying shoreline rotation (Harley et al., 2011; Short and Trembanis, 2004) or smaller-scale and more irregular features such as beach cusps (Coco et al., 1999; Masselink et al., 1997).

## 2.3   OSL

The final step of the GOaL approach is to apply a chronology to barrier formation using detailed morphostratigraphy. Adding a temporal component to coastal formation is important to understand shoreline evolution over timescales that operate on longer terms than that documented historically. Optically stimulated luminescence (OSL) dating is a geochronology technique that determines the time elapsed since buried sand grains were last exposed to sunlight (e.g. Huntley et al., 1985). Upon burial, ionizing radiation from surrounding sediment (by radioactive decay of U, Th, Rb, and K) and cosmic rays is absorbed by the mineral grains and stored in traps within their crystal lattice. Exposure to sunlight can bleach away light-sensitive luminescence signal and reset the "clock" to zero. This stored radiation dose can also be evicted with light stimulation in the laboratory and the energy of photons being released can be measured. Calculating the age when the grain was last exposed to sunlight is based on quantifying both the radiation dose received by a sample since its zeroing event and the dose rate which it has experienced during the burial period. OSL chronology can provide the resolution necessary to decipher the decadal-, centennial-, and millennial-scale patterns of coastal behaviour necessary to reconstruct sea-level curves, determine storm frequencies, and calculate sediment supply and progradation rates.

Originally, dating coastal barrier formations was dependent on sourcing scarce organic matter (often involving extensive coring) and extrapolating the conventional radiocarbon dates from the nearshore to the surface using isochrons (e.g. Fig. 1a). Since OSL chronology determines the time elapsed since mineral grains were buried, this technique dates when paleo-beachfaces and relict foredunes were forming. Dating coastal systems using OSL has been very successful on a global scale (e.g. Jacobs, 2008). Quartz is both a principle mineral used in luminescence dating and abundant in coastal barriers. Therefore, lidar and GPR can be used to target specific stratigraphic layers in a strategic manner for sampling.

This targeted approach using OSL has been shown to more accurately date beach and dune formation than inferred radiocarbon ages from deep nearshore or offshore organic deposits (e.g. Oliver et al., 2015). Oliver et al. (2015) offer an example comparing radiocarbon and luminescence ages at the Moruya Barrier. Because this study focused on comparing chronologies, lidar and GPR data were not presented in Oliver et al. (2015), but both techniques were used to target specific stratigraphic layers prior to OSL sampling (Fig. 1c). The results revised the long-standing theory, based on radiocarbon dates, that the barrier prograded at two different rates before halting 3000 years ago due to diminished sediment supply (Roy and Thom, 1981). The OSL data revealed that the barrier has prograded at a constant rate throughout the Holocene (Fig. 1c). Nevertheless, the radiocarbon dating of shell deposits within the beach facies has been shown to provide similar ages to OSL dates acquired from associated beach and dune deposits (e.g. Hijma et al., 2017; Murray-Wallace et al., 2002). Therefore, where suitable organic material exists within the barrier sands, radiocarbon dating can be utilized at a fraction of the cost and time of OSL analysis. Accurate $^{14}C$ dating requires experience to understand the provenance of the organic material and scrutinize the type of shells used, since beach facies often contain reworked shells (e.g. Rodriguez and Meyer, 2006).

Collection of OSL samples in the field is relatively easy following various methods described in the literature or guidance from someone with experience; however, the processing and analysis of samples requires a scientist trained in luminescence chronology (e.g. Bailey and Arnold, 2006; Huntley et al., 1985; Jacobs, 2008). Therefore, it is not within the scope of this paper to discuss how to process OSL samples (e.g. sample prep and mineral separation) or the complicated intricacies of analysis (e.g. assumptions like water content and burial history or considerations of experimental conditions and statistical models for each sample to be dated). Instead this section focuses on demonstrating the utility of OSL in barrier systems and how it can be optimized by using lidar and GPR to inform researchers on optimal locations for OSL sample collection.

An advantage to OSL dating is that samples can be collected from anywhere in sandy barrier systems in contrast to radiocarbon dating, which relies on organic material often found at depth and thus requires assumptions on the spatial extent of the radiocarbon ages. However, because OSL dating is expensive and time-consuming (relative to radiocarbon), it is advised to strategize sample collection in order to minimize cost while addressing the research questions. The collection and analysis of lidar and GPR provides a detailed understanding of the system stratigraphy, which enables strategic OSL sampling. To ensure accurate sampling of the target facies, it is recommended to also operate the GPR in the field during OSL sample collection, especially if it is collected by coring rather than from an open trench.

When reporting ages in publications, it is important, in particular for younger ages, to indicate the date of OSL measurement (Zhixiong Shen, personal communication, 5 March 2018), since the ages refer to time before OSL measurement. This is also important when presenting both radiocarbon (in calibrated years BP) and OSL dates in the same discussion. While the difference matters little when discussing dates of > 10 000 years, it can be significant for the last 1000 years. For these shorter time periods, converting the radiocarbon ages and presenting all dates in years CE is a

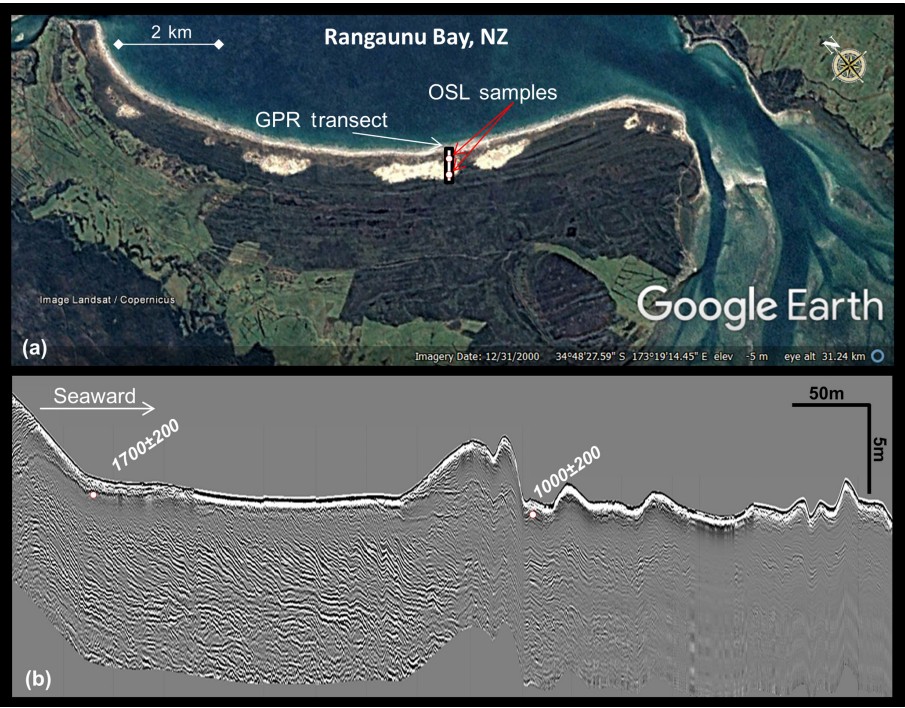

**Figure 5. (a)** Google Earth image of East Beach, New Zealand, and the prograded barrier that it fronts. This aerial image shows the distinct change in morphology from the older vegetated foredune ridges to a large dune blowout fronted by low-lying irregular foredunes with sparse vegetation. This information was used to guide collection of the GPR to image the stratigraphy associated with these two changes in morphology **(a–b)**. The GPR data revealed a major change in the stratigraphy from strong prograded beachface reflections to low-amplitude more chaotic reflections in the beachface. Both the morphology and stratigraphy were used to determine OSL ages (years before 2005) of the youngest intact relict foredune ridge ($\sim$ 1.7 ka) and the timing of the drastic shift in evolution observed in both the dune morphology and beach facies stratigraphy ($\sim$ 1.0 ka). Note that GPR is particularly useful to study nearshore dynamics in the stratigraphy at this site, since its location in the high-energy breaker zone makes this region difficult to access and monitor. Figure modified from Dougherty (2011).

solution (Christopher Hein, personal communication, 14 December 2018).

Morphostratigraphy from aerial imagery, lidar, and GPR is not only useful in determining where best to collect OSL samples that capture uniform progradation to construct complete paleoenvironmental records (Fig. 1), but also to identify significant shifts in barrier evolution so that they can be dated (Fig. 5). East Beach Barrier in New Zealand demonstrates how surface and subsurface data guided OSL to better understand a recent transition from uniform progradation to a more complex evolution (Fig. 5a; Dougherty, 2011). In order to decipher the timing of this shift, aerial imagery was used to target the changes in morphology and GPR to locate corresponding differences in the underlying stratigraphy (Fig. 5). The OSL samples were measured in 2005. The age of the last relict foredune preserved indicates that the barrier prograded uniformly until at least 1700 years ago (Fig. 5b). After this time, a large dune blowout formed, modifying any previously existing morphology. The distinct shift in both stratigraphy and morphology was dated at $\sim$ 1000 years ago (Fig. 5b). This younger age is important to understand the change in evolution within the context of the regional setting, since in

the last millennium three major events could have impacted the coastline: (1) the arrival of the Maori people (Wilmshurst et al., 2008), (2) sea level stopped dropping from a mid-Holocene highstand (Dougherty and Dickson, 2012), and/or (3) a large tsunami struck the area (Nichol et al., 2004).

## 3 GOaL hat-trick (combined GOaL examples)

Recently, three studies have utilized GOaL on prograded systems to (1) reconstruct sea level (Costas et al., 2016), (2) determine the impact of storms (Oliver et al., 2017b), and (3) decipher barrier evolution and sediment supply (Oliver et al., 2017a). These studies are used here as a framework to discuss the significance of GOaL and its potential pitfalls. Where necessary, recommendations are offered in order to improve the robustness of interpretations.

### 3.1 Sea level

Costas et al. (2016) provided an excellent example of how GOaL can be used to reconstruct Holocene sea level from Troia Peninsula, Portugal. Lidar of this complicated spit sys-

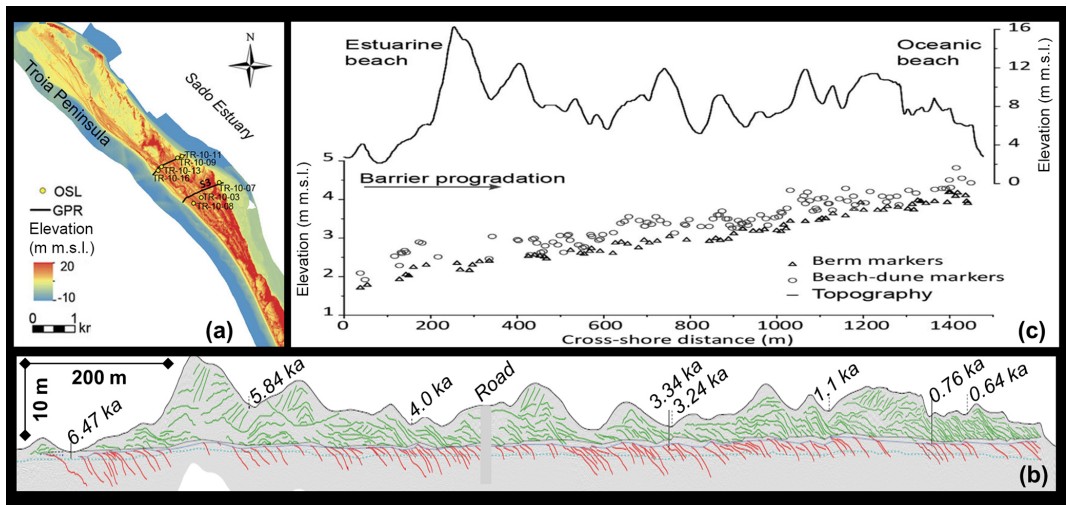

**Figure 6. (a)** Lidar of Troia Peninsula, Portugal, showing locations of GPR and OSL transects. **(b)** GPR transect across the barrier showing interpreted dune stratigraphy in green and beachfaces in red, with associated OSL ages. **(c)** Elevation plot of berm and beach-dune markers used as a sea-level proxy displayed with corresponding overlying dune morphology. Figure modified from Costas et al. (2016).

tem highlights the prograded section of the barrier targeted for GPR and OSL collection across the entire barrier (not one but two transects), capturing a complete progradational history (Fig. 6a). The presentation of both raw (not shown) and interpreted GPR data (Fig. 6b) across the entire barrier is ideal for the reader to see the beach and berm elevational markers used as a sea-level proxy. Complete transects are often not collected for logistical reasons, and when collected, they are often published only partly. It is best to collect at least one single transect line that spans the entire barrier to capture a complete Holocene history. It is also very informative to indicate the location of OSL samples on the GPR profile, regardless of whether it is displayed on the entire record or on selected detailed sections. This allows the specific stratigraphic section dated to be identified. In Costas et al. (2016), topographic profiles of the modern beach and cores were used to ground-truth the GPR such that the berm–beach-dune contact could be interpreted as a proxy for sea level (Fig. 6b); this is summarized in Fig. 6c. Results showed good agreement with known sea-level curves in southwest Europe. This study, along with work from North America (van Heteren et al., 2000; Rodriguez and Meyer, 2006; Billy et al., 2015), demonstrates the potential of applying this method to regions where middle to late Holocene records are not as well documented and/or are debated (e.g. Dougherty, 2018b).

## 3.2   Storms

Oliver et al. (2017b) used GOaL on two proximal prograded barriers (Wonboyn and Boydtown) along the southeast coast of Australia. GPR data spanning millennia to the present-day berms were collected and Oliver et al. (2017b) concluded that all of the paleo-beachfaces in the geophysical record

were stacked post-storm profiles with no berm stratigraphy preserved. However, this interpretation likely overestimates the recurrence and impact of storms due the high gain applied to the GPR data (e.g. Fig. 7b) as well as its presentation highlighting every line (e.g. Fig. 7a) instead of annotating interpreted facies or individual beachfaces (e.g. Fig. 6b). Both representations of the GPR data make it hard to distinguish large-scale facies boundaries (such as a beach–dune interface used for sea-level reconstructions), let alone differentiate storm-eroded from swell-accreted paleo-beachfaces (Fig. 7c–e). For example, in Oliver et al. (2017b) the GPR data on the relict beach–foredune ridges, which are interpreted as only recording high-energy storm conditions, look identical to GPR data on the modern berm, which by its nature was constructed during swell conditions (Dougherty, 2018c). Coring or augering to ground-truth what is causing these strong reflections would have shown the difference between dune and beach facies that are both represented by similar CE2 high-amplitude signatures. Additionally, these cores would have determined which of these strong beachface reflections were caused by erosional lag deposits (e.g. heavy mineral, coarse-grained, and/or shell hash). An ideal implementation of the GOaL approach to extract a regional storm record from prograded barriers is as follows: (1) use lidar to determine the straightest and most continuous transect through each barrier, (2) collect GPR data across the barrier and adjust the gain to highlight the strongest reflections in the beach facies and ground-truth using cores to confirm they represent eroded paleo-beachfaces consisting of storm lag deposits (e.g. Dougherty, 2014, 2018a), and finally (3) use the GPR data to locate the most prominent eroded paleo-beachface reflections and acquire OSL samples from associated materials, preferably the post-storm recovery de-

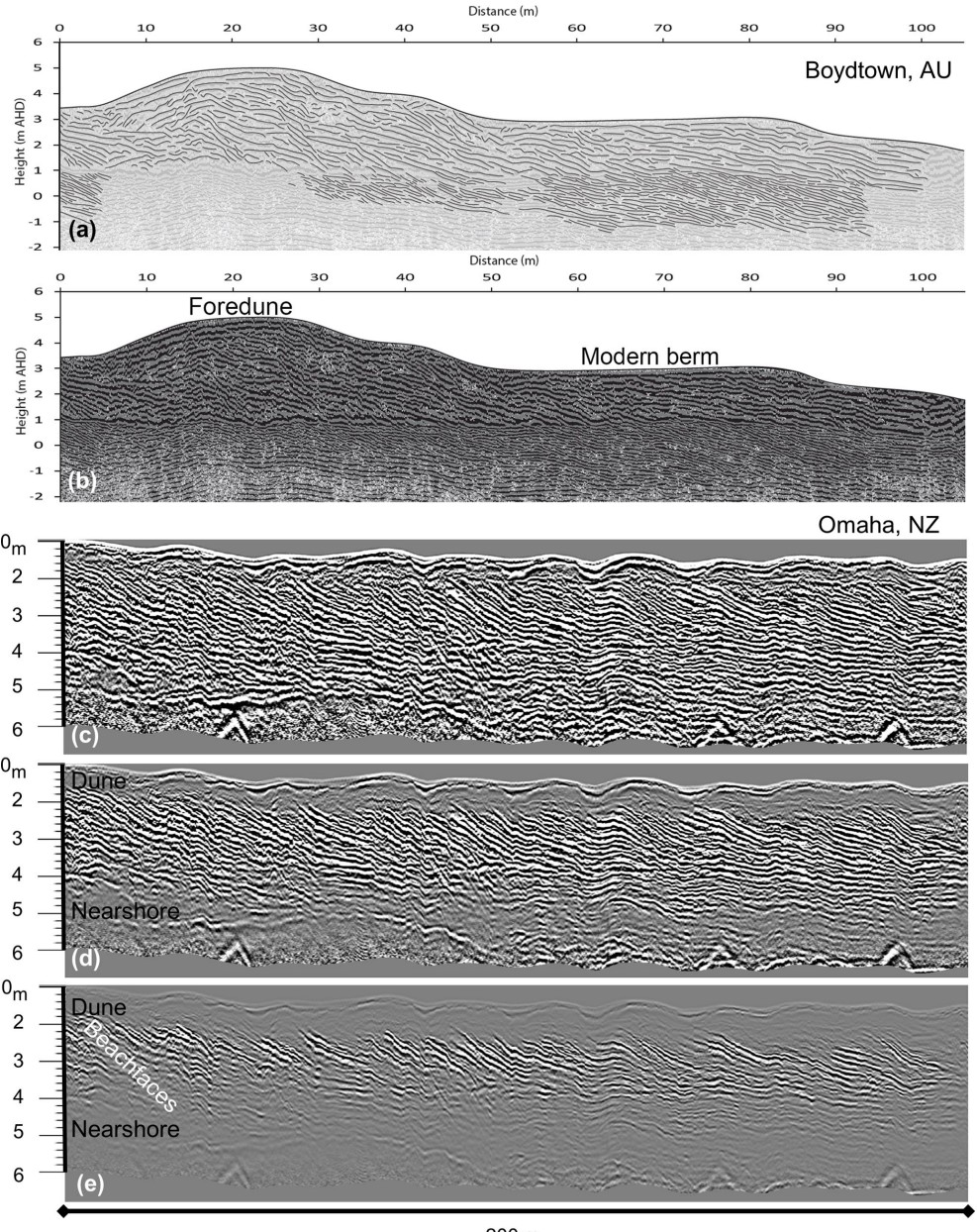

**Figure 7. (a)** An example of the "interpreted" GPR data presented in Oliver et al. (2017b). **(b)** An example of uninterpreted, processed GPR data from the Oliver et al. (2017b) supplementary material showing the high gain applied. Note that in both presentations of GPR data, it is hard to distinguish dune from beach facies, let alone differentiate storm-eroded paleo-beachfaces from the swell-accreted berm stratigraphy. **(c)** GPR data from a prograded barrier in New Zealand (Dougherty, 2014) with a similar high gain applied. **(d)** The same GPR data as in **(c)** but with the gain adjusted so that the more homogenous dune sand is accurately represented as a low-amplitude signal compared to the alternating layers associated with paleo-beachfaces deposited under varying wave energies. **(e)** The same GPR as in **(c)–(d)**, but with the gain control decreased such that the strongest reflections are highlighted. Once these reflections are ground-truthed as high-energy lag deposits, these data can be used to construct a storm record. Figure modified from Oliver et al. (2017b) and Dougherty (2018c).

posit (e.g. Buynevich, 2007). A discussion including these and other studies on proxy records of Holocene storm events in coastal barrier systems is carried out in Goslin and Clemmensen (2017).

## 3.3 Sediment supply and barrier evolution

Oliver et al. (2017a) used GOaL to decipher the complex progradation of Seven Mile Barrier in Tasmania, Australia (Fig. 8a). The GOaL data were used to conclude that changes in sediment supply caused two pauses in progradation be-

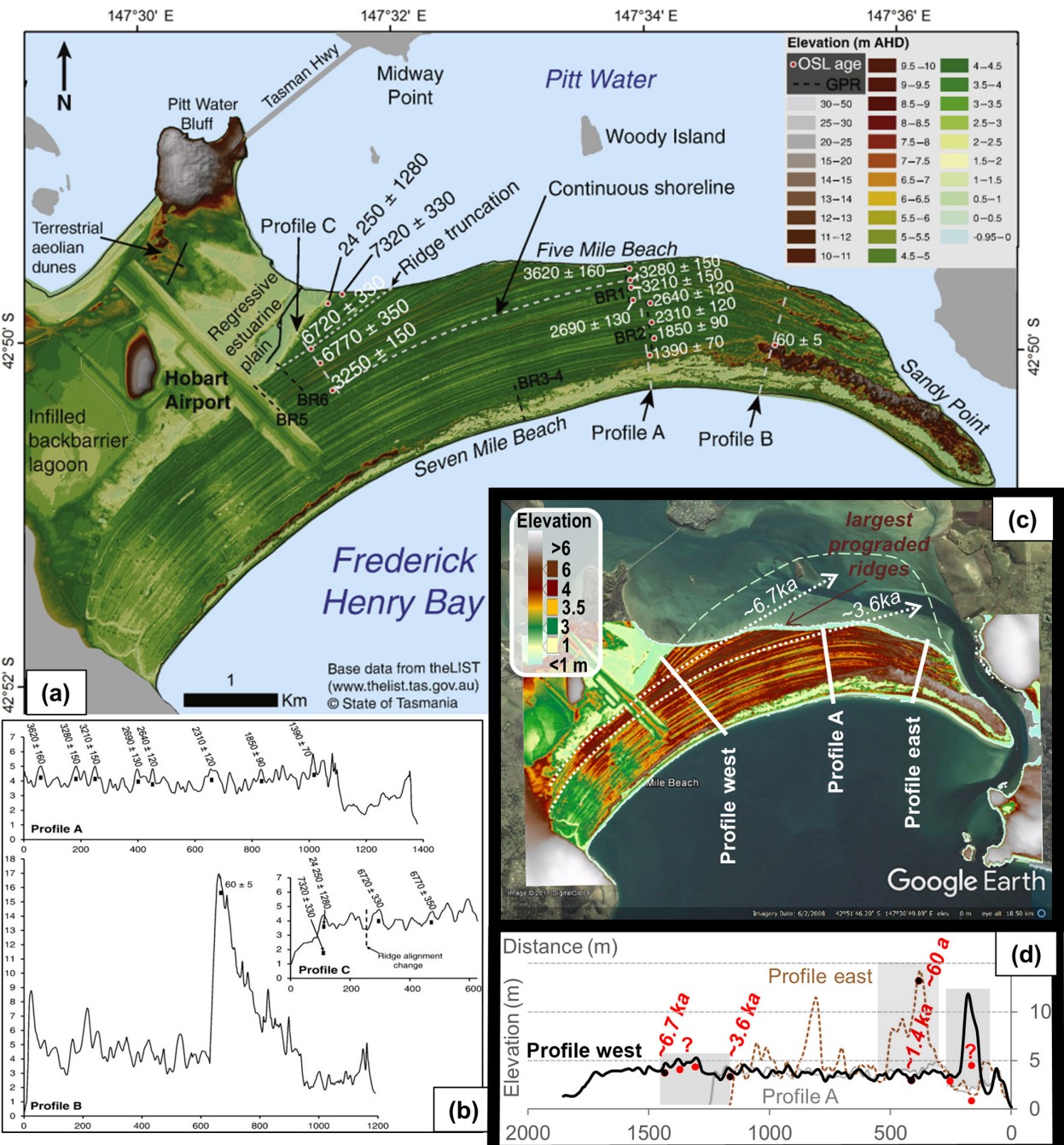

**Figure 8.** Morphology and chronology of Seven Mile Barrier, Tasmania, Australia. **(a)** Lidar data showing the location of topographic profiles in **(b)** and GPR transects (BR no.) as well as OSL ages in years from Oliver et al. (2017a). **(c)** Google Earth image augmented with 5 m lidar (Geoscience Australia; http://www.ga.gov.au/elvis/, last access: June 2017). This combined image shows the potential size of the barrier prior to erosion (dashed green line) and the possible lateral extent of the largest set of prograded foredune ridges which formed between ∼ 6.7 and 3.6 kyr ago (darkest brown ridges within the white dashed lines). **(d)** Topographic profile across the east and west portion of the barrier (location in **c**) overlain on profile A from Oliver et al. (2017a). Profile west transects the entire Holocene barrier, displaying the largest prograded ridges between ∼ 6.7 and 3.6 kyr ago and the extraordinary height of the foredune that formed in the last 500 years, which is missing in **(b)**. Overlaying profile east shows the large 60-year-old dune unconformably deposited above the ∼ 1400-year-old low-lying foredune, indicating relatively recent barrier transgression. Note that gaps in the data coincide with interpreted pauses in progradation by Oliver et al. (2017a), with grey boxes indicating an absence of GPR data and red dots indicating a lack of OSL ages. Also note the vertical age discrepancy in profile C in **(b)** and how GPR could help to understand these age models. Figure modified from Oliver et al. (2017a) and Dougherty (2018b).

tween ∼ 6.5 and 3.5 kyr ago as well as between 500 years ago and the present day (Fig. 8a, b). However, gaps in the morphology, stratigraphy, and chronology coincide with these time frames, raising the question of whether these interpreted hiatuses resulted from a lack of data (Fig. 8d). The breaks in the chronology and stratigraphy may have resulted from how the lidar data were rendered and how topographic profiles were extracted from them. The green colour scheme obscures features in the morphology that distinguish changes in the barrier evolution, such as when the largest relict foredune ridges formed as the barrier prograded a minimum of 200 m between ∼ 6.5 and 3.5 kyr ago (Fig. 8a). To demonstrate the contrast in presentation, 5 m lidar (freely accessible from Geoscience Australia at http://www.ga.gov.au/elvis/, last access: June 2017) was augmented with a Google Earth image (Fig. 8c). This highlights not only the height of these larger relict ridges, but also reveals that these features bifurcate to the east (indicating even greater progradation in this part of the barrier) and extend laterally to the west, filling an abrupt increase in accommodation space (Fig. 8c). The discontinuous topographic profiles presented in Oliver et al. (2017a) not only mask the increased elevation of these prominent relict foredune ridges, but also omit the anomalously large foredune that formed along the southeast half of the barrier over the past 500 years (Fig. 8b, d). While no GPR data were collected for this foredune or the one in the north, evidence of transgression exists as the large 60-year-old foredune is unconformably deposited on top of the 1400-year-old lowlying foredune (Fig. 8c and d). This is an example of how, despite the impressive amount of data that result from combing GOaL, significant features or gaps in data can be overlooked, therefore leading to incorrect or incomplete interpretations. Where it is not feasible to collect parts of the dataset, this absence of data should be acknowledged, addressed, and considered when discussing interpretations or conclusions as well as the level of confidence with which they are asserted.

Modifying the display of existing lidar data and extracting continuous topographic profiles from them identified gaps in the data that challenge the Oliver et al. (2017a) conclusion that sediment supply and progradation paused, with evidence that (1) progradation likely slowed rather than stopped between ∼ 6.5 and 3.5 kyr ago and (2) transitioned to transgression in the last 500 years, suggesting it is unlikely to resume prograding during future sea-level rise (Dougherty, 2018b).

Applying the three-step methodology presented in this paper can optimize the GOaL dataset at Seven Mile. This would not only fill the gap in knowledge with respect to barrier formation and sediment supply, but could also provide insights on the unresolved sea-level record in Tasmania and how its history impacted coastal evolution. An ideal implementation of the GOaL approach at Seven Mile would be as follows: (1) use lidar to identify a transect spanning the entire Holocene record that captures shifts in evolution (western profile in Fig. 8c, d) and utilize areal imagery to locate the nearby road and airstrip that both provide access across

the entire barrier. (2) Collect a continuous shore-normal GPR profile spanning the barrier, with additional data acquired specifically to document the larger foredunes that represent a shift in barrier evolution (Fig. 8d). Then ground-truth the GPR and lidar data using cores, augers, or outcrop mapping on the eroded backside of the barrier, sediment samples, and topographic profiles. (3) Utilize the lidar and GPR to plan OSL sample locations that capture rates of progradation and targets the timing of shifts in barrier evolution that result in the larger foredunes. (4) Integrate the dataset for analysis after all the components have been processed and rendered. (5) Use the digital elevation model from combined lidar and OSL data to calculate the volume of barrier sand above mean sea level to determine sediment budget over time (e.g. Dougherty et al., 2015; Dougherty, 2018b). (6) Combine GPR and OSL data to construct a sea-level curve following published methods (e.g. van Heteren et al., 2000; Billy et al., 2015; Costas et al., 2016). Finally, (7) evaluate barrier formation to determine the nature of shifts in evolution through time and consider them with respect to any changes identified in sediment supply or sea level.

## 4 Concluding remarks

Utilizing GOaL on prograded barriers can provide insights into coastal evolution over spatial and temporal scales spanning from the present-day beach to paleo-beachfaces formed over millennia. Lidar produces 3-D images of the barrier morphology, informing researchers on where best to collect 2-D and 3-D GPR records of dune, beach, and nearshore stratigraphy, which in turn informs researchers on which specific stratigraphic layers should be targeted for OSL dating. In addition to following the simple order to this methodological approach, a few general recommendations can maximize building and interpreting these GOaL datasets: (1) exercise diligence in rendering the lidar dataset and overlay it with aerial imagery, (2) ground-truth the geophysical reflections and apply an appropriate gain control on GPR data, and (3) determine OSL sample locations based on an understanding of the barrier's formation and paleoenvironmental records preserved, then take the lidar and geophysical data in the field (as well as the GPR unit) to locate the targeted stratigraphic layers.

Executing GOaL optimally on a prograded barrier has the potential to generate detailed records of storms, sea level, and sediment supply for that coastline. Obtaining this unprecedented GOaL hat-trick can provide valuable insights into how these three factors influenced past and future barrier evolution. With 300+ prograded barriers worldwide (Scheffers et al., 2012), achieving this GOaL hat-trick systematically on different systems can also detect local patterns of sediment supply, regional records of storms, or global changes in sea level. The prevalence of these coastal deposits and increased accessibility of GOaL techniques affords the possibility to

establish this method such that it can be utilized like and compared with other climate and environmental proxy data. Ultimately, the application of GOaL globally will enable the full exploitation of a precious archive of past coastal evolu-
5 tion and climate change, which in turn will inform practical applications to best mitigate the impacts of global warming on vulnerable communities and infrastructure.

**Data availability.** All data presented in this paper have been previously published with the original sources referenced.

**Author contributions.** AJD conceived the idea for this paper through discussions with the authors and wrote it with input from all of them. AJD contributed over 20 years of expertise in GPR, with incorporation of OSL since 2004 and lidar since 2009. JHC TS1 contributed expertise in OSL. CSMT contributed expertise in cli-
mate change records and suggested the idea to publish this paper as a Technical Note in *Climate of the Past*. AD coined the GOaL acronym and enabled the publication of this paper.

**Competing interests.** The authors declare that they have no conflict of interest.

**Acknowledgements.** We would like to thank Duncan FitzGerald and Ilya Buynevich for sharing their knowledge and enthusiasm for GPR and coastal science. Many thanks to Peter Annan of Sensors and Software for helping customize the use of GPR specifically for this research during a 3-day Pulse EKKO course in Canada,
Mads Toft of Mala GPR Australia for insights gained while trying to get UOW's unit fixed (2011–2013), GBG Australia for offering replacement units and geophysical advice (2006–2013), and everyone at Geophysical Survey Systems, Inc. (GSSI) in New Hampshire for their collaboration and support over the past 20 years. Much
appreciation to John Begg, Navin Juyal, and Vikrant Jain (in New Zealand), as well as Christina Neudorf, Luke Gliganic, Daniela Mueller, Thomas Doyle, Heidi Brown, and Zenobia Jacobs (in Australia) for sharing their expertise in lidar and OSL. Finally, thanks to the editors (Liping Zhou and Denis-Didier Rousseau),
reviewers (Zhixiong Shen and Christopher Hein), and commenters (Marc Hijma and the multiple people that emailed privately) for their contributions to this paper.

Edited by: Denis-Didier Rousseau
Reviewed by: Christopher Hein and Zhixiong Shen

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

**Remarks from the language copy-editor**

CE1     Once again, I'm sorry that this is not in line with your wishes, but lidar is always formatted as such according to our house standards.

CE2     I absolutely understand your hesitation to disturb the editor, but there are simply too many changes here for the proofreading stage, and they need to be approved. Please provide us with a statement to forward to the editor regarding the requested changes. We would send the latest pdf file with the highlighted corrections to the editor for the approval.

**Remarks from the typesetter**

TS1     Please note that we ignore hyphens in names in this section.