# Peer review of "Technical note: Optimizing the utility of combined GPR, OSL, and LiDAR (GOaL) to extract paleoenvironmental records and decipher shoreline evolution"

_Climate of the Past, 2018_

## Short Comment (SC1) · 3 Mar 2018

Nice paper about how the evolution of beach-ridges can be studied. A few general remarks: - The paper focuses on beach ridges, but could easily include chenier plains as well where similar methods are applicable and have been used - With respect to LI-DAR: you could highlight that converting the data to hillshade images and 3D-surfaces (like with ArcScene) is also very helpful - GPR: please inform the reader if it is possible to distinguish between storm surfaces with and without a lot of shells. Do shells have their own specific reflection? - OSL-dating is very useful of course, and I have

used in the Chenier Plain of Louisiana. The existing chronology was based on dating juvenile shells in the 1950's. To my surprise the new OSL-data matched the existing chronology very well, showing that dating juvenile shells was and is still a very useful method and superior to dating scarce organic matter. - OSL-dating should always be done after extensive fieldwork so that is known what facies is being dated. Maybe that could be stressed a little more in the paper. Especially dating aeolian facies is tricky, since it could post data beach-ridge formation significantly. - For OSL-dating is also important to have some idea about groundwater levels since depositing. If the sand was alway below average groundwater level or always above makes quite a difference for the final age - With respect to storm events: dating washover deposits with OSL is also very informative about large storm events. In addition, on chenier plains the washover deposits rest directly on marshland that formed very close to sea level. So at least on chenier plain washover deposits can also be used as sea-level indicators

Section 3.2: it reads a little bit like discussion of a single paper (Olivier et al., 2017b). The authors provide many arguments why some parts of that paper are not so strong. I would prefer a bit more general discussion about dating storm events, including some other papers as well. That would make this section a little bit stronger.

Cheers,

Marc.

---

## Short Comment (SC2) · 5 Mar 2018

This technical note calls for investigating prograded barriers worldwide using a combination of LiDAR, GPR, and OSL (GOaL), with a few general recommendations about how to use and interpret the GOaL dataset. The proposed research strategy to study prograded barrier through LiDAR derived topography informing subsurface GPR stratigraphy collection, which in turn informing geochronology data collection makes perfect sense. The technical note suggests that local to global forcing on coastal evolution can be better deciphered with a large enough prograded barrier dataset collected

following this strategy. This strategy has been applied to earlier coastal studies though not systemically structured like here (e.g., Mallinson et al 2008, Quaternary Research, 69, 97-109). Therefore, it is necessary to call for a systematic and semi-standardized data collection and interpretation as proposed here. One advantage of combining Li-DAR topography and GPR not mentioned yet is that the former is very useful for elevation correction of the latter. However, some common pitfalls of the individual technique are not mentioned, which makes the strategy practically less useful to follow. GPR data collection and interpretation depend not only on gain, but also on the frequency of radar, antenna shielding, spacing of traces, and speed of radar in sediments of different nature. I am not sure why the note specifically picked gain, but not others in the recommendation. OSL age determination is affected by many assumptions about bleaching, distribution of radioactive sources in the sediment, water content variation, postdepositional disturbance, disequilibrium in the uranium and thorium decay series, and cosmic radiation (often a very important component to the total radiation a beach sample received) change because of change of overlying sediment thickness. The choice of appropriate age model does not handle all these complications. One more recommendation about OSL date is that the ages should be reported in a way to enable comparison across different publications. This is because OSL ages refer to the time before OSL measurement and the measurement time must be reported to ensure comparison. As an example of inappropriate reporting OSL data, I noted that the note used 'BP' as a unit for OSL data, which suggests to me that these OSL data refers to AD 1950 following the most common use of BP in the geochronology community. However, my sense is that I am reading the unit 'BP' in the note wrongly.

Specific comments: P4, L4: add 'can' before 'be utilized' P4, L10: add 'of' before 'coastal' toward the end of the line P6, L4: parenthesis for reference not correctly used P6, L12-13: delete 'in the' between the two lines P9, L11: replace 'bleaches' by 'can bleach', and 'any' by 'light-sensitive' P9, L15: replace 'accumulation period' by 'burial period' P11, L5-6: 'In order to decipher the timing of this shift, the aerial imagery was used to target the changes in morphology and GPR to locate corresponding differences

in the underlying stratigraphy'. Can this be shown in Fig. 5b? P11, L14-15: repeating reference P11, L20-21: 'Initially this complicated spit system did not appear as an ideal site to extract a sea level history'. What is the reason for this? P14, L7: sentence toward the end does not seem to be complete P16, L4: replace 'form' by 'from' P16, L5: replace 'intern' by 'in turn' Fig 1: the OSL data seem represented by circles filled by brown, but not open black circles as indicated in the caption. Fig 2: legend in 2b is not legible Fig 6: the thin dashed lines in 6b are not interpreted. Why are there two different y-scales in 6c? What is the difference between berm markers and beach-dune markers? Are the latter beach/dune boundary?

---

## Author Comment (AC1) · 7 Mar 2018

Hi Marc Hijma,

Glad you thought the paper was nice and thank you for taking the time to comment. In order to be sure that I address all of your feedback I will go through your remarks one by one below:

The paper focuses on beach ridges, but could easily include chenier plains as well where similar methods are applicable and have been used

[Figure]

-Indeed, the GOaL method could similarly be applied to cheniers. In fact, I think this approach could be used to study the geomorphology and landscape evolution in a lot of different geologic settings. Also, in cheniers I would think that radiocarbon dating would be the preferred technique.

With respect to LIDAR: you could highlight that converting the data to hillshade images and 3D-surfaces (like with ArcScene) is also very helpful

-This is very impressive and can be useful in detecting and analyzing features. While it can help to visualize the data I have not found that it identified something that was missed in 2D and can be hard to display in print.

GPR: please inform the reader if it is possible to distinguish between storm surfaces with and without a lot of shells. Do shells have their own specific reflection?

-We have just worked on the first sandy system that actually has what I would call a lot of shell concentrations along individual beachfaces. I haven't been able to process the data in detail, but it didn't initially look any differently than other systems with little to no shell preserved in the beachface. Within the systems that I have worked on where there was shells and shell hash concentrated on the lower beachface, the signal looked similar to its upper beachface component that was composed solely of heavy minerals. Shells can sometime cause ringing (hyperbola) in the data, but if you are specifically interesting I would always ground-truth with an auger or core.

OSL-dating is very useful of course, and I have used in the Chenier Plain of Louisiana. The existing chronology was based on dating juvenile shells in the 1950's. To my surprise the new OSL-data matched the existing chronology very well, showing that dating juvenile shells was and is still a very useful method and superior to dating scarce organic matter.

-I have also found that OSL ages are similar to radiocarbon dates of shells within the beach or dune facies; however, there is a discrepancy between the ages when the

organic matter dated is from the near shore or deeper. So yes if you can find shells in the beach facies, getting ages is probably easier, faster, and cheaper than using OSL. It is just that organic matter is often scarce and unless you have shells in the beachface throughout your barrier it might not be possible to date the specific features of interest. OSL on the other hand offers the possibility of dating the formation of any feature in the sandy system.

OSL-dating should always be done after extensive fieldwork so that is known what facies is being dated. Maybe that could be stressed a little more in the paper. Especially dating aeolian facies is tricky, since it could post data beach-ridge formation significantly.

-Agreed, this is why it is the final step in the methodology after the morphology from LiDAR has informed the GPR collection and the stratigraphy from the GPR has been ground-truthed with cores. At this point the evolution should be well understood and then chronology added to either date changes identified or collect a suite to identify a more consistent progradation. Also because of the accuracy of GPR it should he fairly easy to locate the approximate location of your OSL sample within geophysical stratigraphy and I advocate showing that location on the geophysical records. Personally, I take a printed copy of the processed GPR in the field, along with the GPR unit, to locate the exact layer I have targeted to sample.

For OSL-dating is also important to have some idea about groundwater levels since depositing. If the sand was always below average groundwater level or always above makes quite a difference for the final age

-True, this is an important consideration for incorporation.

With respect to storm events: dating washover deposits with OSL is also very informative about large storm events. In addition, on chenier plains the washover deposits rest directly on marshland that formed very close to sea level. So at least on chenier plain washover deposits can also be used as sea-level indicators

-Yes, washover deposits and OSL are informative about large storm events, but in systems where sediment supply great enough to causes progradation and a series of large dunes these features are not common. Since this paper focuses on prograded barriers that lack washovers a discussion of them is therefore not included. Again, with respect to cheniers there is great potential to extract sea level and storm histories that can be compared to sandy prograded systems to better understand paleoenvironmental factors and past coastal evolution. While I have done this in the past (Dougherty, 2011; Dougherty and Dickson; 2012; Dougherty, 2014), I have not had the opportunity to work extensively in cheniers as they are less common features. I am also not aware of a paper where GPR, OSL and LiDAR have all been used on a chenier plain, but if you know of one please share. Regardless, cheniers are not the focus of this paper and while there is much to be learned by comparing and contrasting these systems with proximal sandy barriers, their evolutionary differences are great enough that they were not included in this paper for the sake of being concise.

Section 3.2: it reads a little bit like discussion of a single paper (Olivier et al., 2017b). The authors provide many arguments why some parts of that paper are not so strong. I would prefer a bit more general discussion about dating storm events, including some other papers as well. That would make this section a little bit stronger.

-There are a few papers that discuss detecting and dating storm events, I cite some examples that have done this successfully using one or two of these techniques in the intro and methods (e.g. Buynevich et al., 2007; Buynevich et al., 2004; Dougherty et al., 2004; Dougherty, 2014; Nott and Hayne, 2001). A nice paper on this subject, that includes some of these references, was published last year in Earth Science Reviews (v.174: p.80-119) by Jerome Goslin and Lars B. Clemmensen entitled "Proxy records of Holocene storm events in coastal barrier systems: Storm-wave induced markers". I had meant to add this reference to this paper before submitting, but already have it in the next draft. Along these lines I offer examples in the introduction that utilize GPR, OSL and LiDAR on prograded barriers, independently or in various combinations, that

have been able to: "(1) quantify frequency-intensity of storm records (e.g. Buynevich et al., 2007; Dougherty, 2014; Nott and Hayne, 2001), (2) construct sea-level curves (e.g. Nielsen et al., 2017; Rodriguez and Meyer, 2006; van Heteren et al., 2000), (3) calculate sediment budgets (Bristow and Pucillo, 2006; Dougherty et al., 2015; van Heteren et al., 1996), and (4) decipher coastal evolution (e.g. Barboza et al., 2009; Costas and FitzGerald, 2011; Hein et al., 2016)." These would be just a few suggestions of papers I recommend referencing for more particulars about acquiring similar records. The aim of Section 3 is not a discussion of how to do this, as it has already been done before. Rather, Section 3 is a discussion of three different papers that all came out in the previous year that combined GPR, OSL and LiDAR, each specifically to address one of the three aims of GOaL: sea level, storms and sediment supply/evolution. The critique highlights strengths and address common pitfalls so that future studies using GOaL can be optimized. Based on you comment I will reincorporate the appropriate references from the introduction into the end of the appropriate subsection of Section 3 to remind people where to look for recommended examples.

Hopefully that provided the insight you were seeking. Cheers, Amy

---

## Author Comment (AC2) · 7 Mar 2018

Hi Zhixiong Shen,

Thank you for your comments on this technical note. Glad to hear you think GOaL makes perfect sense and see the potential of having a more systematic and semi-standardized data collection as well as interpretation. I also appreciate the specific comments and the reminder to reference the good work by Mallinson and co. in some of my examples. In order to be sure that I address all of your other feedback, I will go

through your remarks one by one below:

One advantage of combining Li-DAR topography and GPR not mentioned yet is that the former is very useful for elevation correction of the latter. However, some common pitfalls of the individual technique are not mentioned, which makes the strategy practically less useful to follow.

-LiDAR is useful for rough topographic correction. Due to the fact that I use most of my data to try and infer sea level from heights of beachfaces and storms form their geometry, I like to use the most accurate topographic correction possible. To eliminate the uncertainties associated with LiDAR and extracting the exact transect line form the image that the GPR was collected along (usually the two data sets are acquired on different dates allowing possible discrepancies), I prefer the old school method of levelling or laser levelling in the actual elevations of the GPR transect line. This should be done at the time of collection and include a survey of the active beach and tie in any existing benchmarks. This is standard along with coring to ground-truth depth to water table on the day of GPR collection.

GPR data collection and interpretation depend not only on gain, but also on the frequency of radar, antenna shielding, spacing of traces, and speed of radar in sediments of different nature. I am not sure why the note specifically picked gain, but not others in the recommendation.

-True there are many different set up and settings when it comes to collecting GPR. What is ultimately chosen depends greatly on the access to gear, the field conditions, the scientific question to be answered, etc. At this point there are plenty of papers and books about GPR that people can turn to in order to understand how GPR works and what settings are best for each particular study. The aim of this paper was to show the potential of optimizing the use of GPR in combination with OSL and LiDAR, not a' how-to' guide with specifics for acquiring and analysing each data set. Rather this paper works on the basis that readers have a standard knowledge of, and some

experience with, these techniques. As advocated in the paper, it is important to lean from or collaborate with experts using each of these three methods as they are specific fields within geophysics, geochronology and remote sensing.

-With respect to GPR, the units have become so affordable and user-friendly it has become relatively easy to access a machine, turn it on, and get data. While this is absolutely fantastic for the expansion of its use in many fields, there is a danger that without proper knowledge and training data can be collected incorrectly and/or misinterpreted. For the most part in the field of coastal research this is not the case, probably because it has been used in this field for so long with many people and papers to turn to. Therefore, most papers have the basics correct. The reason gain is singled out is for two reasons: 1) applying a high gain (as well as highlighting every reflection) is one very common occurrence in coastal data and 2) it is an incredibly important adjustment when trying to extract storm and sea level records from sandy barriers. I have increasingly noticed that along with the proliferation of GPR use, more papers and conference presentations don't have the gain adjusted so that the signal strength aptly reflects the contrast in the stratigraphy. I have seen high gain inhibit the identification and extraction of a sea level curve from otherwise good data and result in an exaggeration of the storm frequency and impact as demonstrated in Oliver et al. (2017b).

OSL age determination is affected by many assumptions about bleaching, distribution of radioactive sources in the sediment, water content variation, post depositional disturbance, disequilibrium in the uranium and thorium decay series, and cosmic radiation (often a very important component to the total radiation a beach sample received) change because of change of overlying sediment thickness. The choice of appropriate age model does not handle all these complications. -These are important considerations and will be assessed for incorporation within the manuscript.

One more recommendation about OSL date is that the ages should be reported in a way to enable comparison across different publications. This is because OSL ages refer to the time before OSL measurement and the measurement time must be reported

to ensure comparison. As an example of inappropriate reporting OSL data, I noted that the note used 'BP' as a unit for OSL data, which suggests to me that these OSL data refers to AD 1950 following the most common use of BP in the geochronology community. However, my sense is that I am reading the unit 'BP' in the note wrongly.

-Another good point, this will be addressed in the final draft of the manuscript if accepted.

Hopefully that provided the insight you were seeking. Cheers, Amy

---

## Referee Comment (RC1) · C. Hein (Referee) · 19 Mar 2018

General Comments:

Recent years have been exciting for research on progradational systems. It is during this time that scientists have been moving at an accelerated pace from the study of the evolution of beach ridges (and associated landforms) as geomorphic features to the use of these progradational systems as archives of past changes in sea level, sediment supply, wave energy, and storms, among others. These authors present some excel-

lent thoughts and summarize several best-practices for using and coupling different topographic, subsurface, and geochronological datasets to achieve these latter goals. This manuscript presents a case for the integration of LiDAR-based topographic data, ground-penetrating radar (GPR) -based stratigraphic data, and optically stimulated luminescence (OSL) -based geochronologic data in reconstructing paleo-environmental change (storms, sea-level, etc) from beach- and foredune- ridge plains.

The manuscript is well written, well motivated, and very well organized, particularly the inclusion of general best practices for each of the tools, followed by discussion of a series of case studies which have successfully applied this approach. The basic premise is indisputable: the combination of multiple tools can allow for high-resolution mapping as well as comprehensive reconstruction of system evolution, and in particular those components which could impact paleoenvironmental reconstruction interpretations. Not following these, and other, best practices in the study of these clastic depositional systems can lead to misinterpretation of the paleoenvironmental record.

I suggest four general revisions for this manuscript: 1) Consider the addition of ground-truthing as a fourth approach, equally as important as LiDAR, OSL, and GPR. 2) Recognize the limitations of certain field sites and conditions which may make any one of the three (or four) "hat trick" components not possible, or not the best approach for a given site. 3) Consider adding examples from additional global sites; this does not need to be in the discussion of the three "case studies" as those are meant to be focused on single papers. 4) consider a more measured treatment of the Oliver et al (2017a,b) studies. I elaborate on each of these below, with specific examples and suggestions.

General Revision #1: I would argue that the "hat trick" described (LiDAR, GPR, OSL) might be better constructed as a "grand slam" (are we taking the sports analogy too far?). That is, sediment cores (or some other ground-truthing of non-invasive subsurface data) are a critical fourth component to these types of studies, equal in importance to remote stratigraphic data, high-resolution topography, and proper dating.

[Figure]

General Revision #2: The "hat trick" goal is not always achievable, as the authors recognize. Despite advances in data collection using balloons, kites, drones, as well as similar topographic mapping techniques using ground-based RTK-GPS mapping and aerial structure-from-motion, collection of high-resolution data over large areas (e.g., complete beach- and foredune- ridge plains) remains expensive. The best datasets tend to be those collected by government agencies and made publically available; while the accessibility of such databases (and other tools for mapping by individual scientists) is rapidly expanding, they are far from ubiquitous, especially in remote locations. Likewise, GPR suffers from signal attenuation by saltwater and muddy sediments; the latter is a particular challenge in trying to integrate stratigraphic data from modern beach systems. Finally, OSL is geographically limited to systems composed of quartz sand and, as commenter Shen notes, subject to methodological assumptions, error, and reporting inconsistencies. Other chronological techniques such as radiocarbon dating of (ideally in situ) shells, remain generally less expensive and applicable to coastal systems composed of, for example, mud, coarse gravel, or carbonate sand. For example, Long et al (2012; QSR 48:61-66) present a robust sea-level reconstruction from coarse clastic beach ridges in Svalbard based on bivalve 14C dating. Billy et al (2015; Geomorphology 248:134-146) do the same using GPR and RTK-GPS-based topographic mapping, but with OSL largely from sandy foredune ridges which overlie (and therefore may postdate by some unknown amount of time) a coarse clastic beach-ridge plain which could not be sampled for OSL. Thus, while the authors present what may be an "ideal" approach, it is important to recognize that site or data limitations do not invalidate studies that cannot achieve the GOaL "hat trick". In particular, radiocarbon should certainly not be dismissed out of hand for all sample types and depositional environments.

General Revision #3: As this is a "technical note" and not a comprehensive review, it is understandable that it derives examples primarily from the authors' study sites. However, the focus on Australia and New Zealand leaves out possible lessons some recent examples of comprehensive ("GOaL") studies of beach-ridge plains; see, for

example, Clemmensen & Nielsen, 2010 [Sedimentary Geology 223:281-290], Hede et al 2015 [The Holocene 25:1402-1414], and Nooren et al 2017 (Earth Surface Dynamics Discussions). That said, the authors do present a very recent case study from Portugal (Costas et al. 2016) of a sea-level reconstruction using this "hat trick" of approaches which they deem to be of highest quality (I agree that is a very robust study).

General Revision #4: The authors present GOaL studies of storm records and sediment supply from Oliver et al (2017a,b), from which Dougherty et al. draw inferences about missed opportunities to use some of the best practices described. Included in this is a comment (P12, L19-21) that ground-truthing with sediment cores would have improved interpretations, unintentionally bolstering my suggestion for a "grand slam" approach! While Dougherty et al make a strong case for additional data that could have been collected, analyses that could have been done, and several alternative (although not necessary superior) interpretations, I suggest a more measured approach that recognizes the creative approaches and robust data presented by Oliver et al. Here I elaborate further, reflecting the "interactive discussion" nature of the "Climates of the Past" review process.

The authors argue that the records from the barriers presented by Oliver et al (2017a,b) are incomplete. There is absolutely a case to be made for record (storms, sea-level change, etc) reproducibility within a single plain. Ideally any such record would integrate two or more parallel lines of radar, topographic, and chronological data extending across the entire plains; indeed this was a strength of the Costas et al (2016) study. However, the reality of field studies is a limitation of these studies. For example, rarely are large progradational plains preserved without later erosion or reworking by wind or water. Whereas a tidal creek has eroded/modified part of the Boydtown barrier (Dougherty et al, P13, L4-5) studied by Oliver et al 2017 (Geomorph), imagery published in that paper suggests that at least parts of every beach ridge are preserved (although, we all must recognize the possibility of erosion of one or more complete ridges entirely alongshore soon after deposition); Oliver et al appear to have sampled

nearly every ridge across that plain. However, this identifies a second nearly ubiquitous issue with the study of progradational systems: although "natural" (undeveloped) plains best preserve natural topography, that same topography (and associated vegetation) can negatively impact the quality of GPR data, or render sections of the study area inaccessible. The latter appears to have impacted the studies of both Wonboyn and Boydtown. Nonetheless, although Dougherty et al make the important point that reproducibility (multiple, parallel plain-wide reconstructions) would surely have strengthened that record, the resulting storm record is not invalidated, as Dougherty et al. seem to suggest.

In this same vein, the Dougherty et al. (section 3.3) rightfully note that the integration of multiple, parallel, cross-shore topographic profiles from Seven Mile Barrier would have improved, and possibly altered, interpretations by Oliver et al 2017 (MarGeo). However, even from the data presented by Dougherty et al., the interpretation regarding the sediment fluxes between 6700 and 3600 years ago (P14, L12-14) would only subtly change: progradation/deposition perhaps did not "pause" or "halt" as interpreted by Oliver et al, but most certainly slowed quite significantly. Indeed, the vertical growth of foredune ridges can be interpreted as a slowing of progradation, allowing more time for sand to be transferred into beach-adjacent ridges (see, e.g., Nooren et al 2017, cited above). Likewise, for the modern system: Dougherty et al identify a substantially higher modern foredune ridge, which in the west of the plain appears to be transgressive in origin. This too suggests a decrease in sediment flux, or at minimum, a sediment flux unable to keep pace with accommodation creation (sea-level rise). The implication of this observation for future growth of this plain (P.14, L21-22) differs from the more conservative observation by Oliver et al., but perhaps not as substantially as suggested.

Thus, Dougherty et al identify some site-specific issues with studies by Oliver et al (2017a,b), which may have led to mis-/over-interpretation of data. Most importantly, Dougherty et al identify several transferrable lessons for the use of the GOaL techniques. However, these findings seem to impact the subtleties of the Oliver et al

(2017a,b) interpretations; their overall conclusions about system evolution or large-scale records contained within appear to remain robust. Oliver et al (2017a,b) push the envelope with creative uses of progradational records, and did well to integrate modern and Holocene geomorphic data and use the GOaL datasets creatively to investigate sediment fluxes and storms. Thus, while Dougherty et al make some excellent and worthy points, they might consider a more measured treatment of these two published studies, and focusing this section of the manuscript on the lessons for best practices.

Specific Comments

P6, L9 & Figure 2: this is focused on the utility of LiDAR, not the details of this study. However, it is not clear that the multiple sets of "prograded barrier islands" shown here were never a single island / beach-ridge plain. This is a great example of possible reworking of a non-continuous record, a limitation in reconstructing the evolutionary history of a progradational site, or the paleoenvironmental records contained within. This in fact may be a case for the use of subsurface data (GPR) to search for, e.g., landward-dipping beds at the landward side of each of those "islands" to try to infer if they formed as separate transgressive-regressive islands. LiDAR data here may in fact tell an incomplete story.

P8, L8-9: this statement would be stronger if supported with examples or details of how beachface mapping can be used to infer sea level, etc.

P8, L14: suggest being more specific. What about the change in reflection geometry indicates storms?

P8, L19-26: If the authors are going to discuss these aspects of GPR processing and interpretation (including the necessity of ground-truthing, as discussed earlier), then it is also worth noting some additional key processing steps for proper GPR interpretation. For example, migration, ideally using field common-midpoint (CMP) surveys, to determine radar velocities.

P10, L10: "calculations from the LiDAR data". It may be worth noting that this would not have been possible without ample stratigraphic data from sediment cores, especially given the limited GPR penetration.

P12, L15: "gain control is high in the GPR data". That could just be the way in which the GPR data are shown in published form; that can be a challenge to get right. Presumably the GPR data were analyzed in high detail, and through careful tuning of gains to ensure best analysis resolution. Only those authors can speak to this. This same interpretation could be applied to criticisms of Oliver et al (2017a) noted on P14, L7-8 (LiDAR color scheme chosen for publication display).

Technical Corrections

P2, L20-21: the point concerning collaboration between scientists with specific expertise in each of the tools (some of which [e.g., GPR or pre-processed LiDAR data] are perhaps easy to use, but not easy to use well!) described in an important one.

Figure 5 caption: "prograded normally for a while". This is unclear, unspecific, and qualitative. (the term "normally" is applied on P11, L7 as well, and does not seem to necessarily indicate "normal [sediment surplus driven] progradation". Correct?). "drastic shift in evolution observed . . ." this is not clear from the data presented, nor is it clear what would qualify as a "drastic" change

P11, L15: Costas et al 2016 is listed twice

---

## Editor Comment (EC1) · L. Zhou (Editor) · 4 Jun 2018

Dear Amy, Would you like to submit a Reply to the comments made by Referee 1 (RC1), which was posted on 19 March? Liping Zhou (lpzhou@pku.edu.cn)
* * *

---

## Author Comment (AC3) · 5 Jun 2018

Dear Professor Zhou, I would like to submit a reply to Referee 1, but was waiting for the second review. I just received an email Friday from Natascha Töpfer explaining the confusion and delay. I presume that my previous reply to the short comment/1st review is sufficient and will now complete the reply to the second review (RC1) ASAP. Cheers, Amy

---

## Author Comment (AC4) · 30 Jun 2018

The authors would sincerely thank Chris Hein for his thoughtful and thorough review. These comments, along with others posted online and emailed privately, will help hone an improved manuscript should it be accepted for revision.

In following with Climate of the Past's guidelines, this response will be structured such that each revision/comment will be addressed in numerical order following the sequence: a) comments from referee, b) author's response, and c) author's changes

in manuscript.

General Revision #1: a) Consider the addition of groundtruthing as a fourth approach, equally as important as LiDAR, OSL, and GPR. b) While the idea of a grand slam is an appealing one, groundtruthing is not seen as a standalone technique per se. Rather coring or topographic profile collection (for example) as a means of groundtruthing remotely sensed data, is seen as an integral component of GPR and LiDAR methods. While LiDAR, GPR, and OSL can all be used individually or in various combination, it is not recommended that GPR or LiDAR is used without being groundtruthed (or at the very least state the omission and consider when interpreting the data). Coring or outcrop mapping used to ground-truth GPR and topographic profiling using levels, lasers, or GPS to ground-truth LiDAR are all techniques that can be used alone or in combination with other methods (e.g. air photograph analysis or radiocarbon dating) to study coastlines. Ultimately, our counter argument would be that groundtruthing of non-invasive subsurface data is not a critical fourth component to GOaL approach, but rather an essential element whenever remotely sensed high-resolution stratigraphic or topographic data is used (therefore embedded in GPR and LiDAR methods). c) No changes are planned for incorporating groundtruthing as a fourth approach. The point will be clarified about groundtruthing remotely sensed data is not just mandatory for the combined GOaL approach but whenever GPR or LiDAR is used.

General Revision #2: a) Recognize the limitations of certain field sites and conditions which may make any one of the three (or four) "hat trick" components not possible, or not the best approach for a given site. b) It is recognized that the combination of GPR, OSL and LiDAR is not always the best approach or even possible for a given site. However, for those sites where these techniques are able to be used this paper aims to provide insight on how to optimize their utility. We are not insinuating, nor state within the paper, that this is an "ideal" approach for all sites. What is inherent, but seems to be made more explicit, is that the optimizing of this GOaL approach was conceived for sandy prograded barriers. Following on from the discussion of the previous comment the use of GPR, OSL, and LiDAR does not preclude the use of other methods and in some cases necessitates it. Of course where there is organic material for radiocarbon dating, use that instead of or with OSL. c) Having said that, we will include the limitations of GPR, OSL and LiDAR suggested by the reviewer. We will also make clear that this is approach and discussion is geared toward sandy prograded barriers that are meant to complement other studies using various techniques to document the morphology, stratigraphy and chronology of other coastal systems (such as the examples provided by the reviewer: Long et al (2012; QSR 48:61-66) and Billy et al (2015; Geomorphology 248:134-146). This will also include mention of chenier plains as discussed in the response to the comments by Marc Hijma. With regard to the expense of the GOaL techniques, it is agreed that they are still costly. However, the price has come down and access has increased rapidly as of late. This is likely to continue into the future and the aim of this paper is to suggest a few basics moving forward that can optimize the combined use of these methods. General Revision #3: a) Consider adding examples from additional global sites; this does not need to be in the discussion of the three "case studies" as those are meant to be focused on single papers. b) This is a very good suggestion that does indeed broaden and strengthen the paper. In addition to some of references that were meant to be included, the response to this interactive discussion (both in the form of online comments and private emails) provided suggestions for other papers that will also be included. It is true that when drafting the paper the idea was to use the three most recent examples that use GOaL specifically to study sea level, storms, and sediment supply as case studies. However, along the lines of Marc Hijma comment, it will be nice to reiterate some previous examples of successful studies mentioned earlier in the paper as reference when individually discussing study sites specifically on sea level, storms, and sediment supply. c) Additional references (such as those suggested in this review as well as the comment by Zhixiong Shen) will be included in the paper from sites spanning the globe.

General Revision #4: a) Consider a more measured treatment of the Oliver et al (2017a,b) studies. b) Since the submission of this manuscript to Climate of the Past, an

extensive comment on Oliver et al. (2017a) has been published and a similarly detailed discussion paper on Oliver et al. (2017b) has been accepted with minor modifications (Dougherty, 2018a,b). These comment papers allow a nuanced conversation of the data and interpretations presented in each that was simply not possible nor appropriate for this technical note. This allows us to refer to these comment papers for a fuller discussion when identifying the potential pitfalls encountered when GPR, OSL and LiDAR are not used optimally. This technical not will focus specifically on how interpretations can be questioned when GPR data is not groundtruthed with cores as well as when rendering of LiDAR (and topographic profiles extracted from the remotely sensed data) masks or distracts from important aspects of the morphology. The importance of groundtruthing is agreed, as per the discussion above, and presnetaion publicly does matter if it is influencing interpretation (see discussion in Specific Comment #6 below). While it is uncomfortable to critique specific studies, it is necessary in order to have a rigorous scientific debate. Because of the critical nature of this aspect of the paper, it was important to us that it was published in a way that the authors of the Oliver et al. (2017a, b) papers could comment or correct anything that might be misleading. We are grateful for Climate of the Past's open access and interactive review process which allowed this option. Authors associated with both papers were made aware of this technical note addressing methods used in their papers before it was submitted and it is known that the lead author has viewed this Climate of the Past submission (as informed by ResearchGate). c) The two sections referring to the Oliver et al. (2017a,b) studies will be redrafted so that specifics to the study sites are minimized. Many of the points raised by the reviewer are addressed with greater context in the comment papers and since they would not be included in a revised draft of this paper if accepted, we will not reply to specific aspects here. Rather outline the main points that will be addressed in the revision of these two sections and points raised by the reviewer that pertains to them. Ultimately, trying to focus these sections on the transferrable lessons for the best practice use of the GOaL techniques (as suggested by the reviewer).

3.2 Storms To determine a storm record requires eroded paleo-beachfaces to be

clearly identified within the stratigraphy. This requires coring of these storm layers and using these cores to groundtruth the GPR and make gain adjustments. Failure to do this can result in ambiguous interpretation of storm plaeo-beachfaces. A new example of GPR data will be added that shows GPR data with high gain applied masking the storm layers. This same data with the gain reduced according to an overlain core reveals obvious storm layers. This will demonstrate the potential to interpret an over exaggerated storm history for sites that do not core or adjust gain, such as Oliver et al. (2017b).

3.3 Sediment supply and coastal evolution In order to discuss changes in evolution through time and not confuse these with alongshore variations, a single shore-perpendicular transect line from the oldest to the youngest part of the barrier is optimal. This GOaL paper advocates for using LiDAR to determine where the best location is to collect this transect. Then collect GPR along this transect and use this morphostratigraphy to target the best location to collect OSL. We agree with the reviewer that integrating two or more parallel lines improves the robustness as well as allowing a discussion about alongshore variation through time. However, this is not our point or the reason for the multiple lines drawn in Figure 7d. Rather these lines were extracted from the rerendered LiDAR to demonstrate how it can be used to determine the best location to extract the best transect to span the entire history. This process clearly shows that the western profile is the most complete. Comparing this one transect to those presented by Oliver et al. (2017a) identifies gaps within the morphologic data at points where the evolution shifts towards the beginning of barrier inception and in the most recent period. The LiDAR shows the accommodation space changes through time and identifies that anomalously large foredune ridges formed during this time. However, GPR and OSL were not collected for these areas of interest. The gaps in the topographic record, GPR stratigraphy and OSL chronology during these two points in time raises the question of whether the interpretation of halted progradation is due to ceased sediment supply or just a result of a lack of data. At the very least identifying that these gaps exist and considering the implication is crucial to any discussion about barrier

evolution and the role of sediment supply. This does not negate sizable amount of data presented for the other area of the work that no doubt went into collecting it. Rather that following the simple order and suggestions presented in this GOaL methodology might have helped optimize this dataset by first targeting shifts in evolution using LiDAR morphology, detail how these sections of the barrier formed using detailed stratigraphy from GPR and then dating the timing of these changes using targeted OSL samples.

Specific Comments #1: a) P6, L9 & Figure 2: this is focused on the utility of LiDAR, not the details of this study. However, it is not clear that the multiple sets of "prograded barrier islands" shown here were never a single island / beach-ridge plain. This is a great example of possible reworking of a non-continuous record, a limitation in reconstructing the evolutionary history of a progradational site, or the paleoenvironmental records contained within. This in fact may be a case for the use of subsurface data (GPR) to search for, e.g., landward-dipping beds at the landward side of each of those "islands" to try to infer if they formed as separate transgressive-regressive islands. LiDAR data here may in fact tell an incomplete story. b) The main point of this section was to demonstrate how LiDAR can provide improved images of the morphology of coastal system and their surroundings to inform where to collect GPR. This approach was used at Rangitaiki Plains, but the GPR aspect was excluded so that LiDAR could be the focus. Based on this comment, it seems pertinent to add that this informed GPR collection. The combination of these data, along with previous studies of the area, was used to determine the evolution of this complex coastal system. The evidence for this series of prograded barrier islands is that the landward extent of each naturally transition into back-barrier deposits. These cohesive muds were more resistant to erosion and therefore preserved the morphology of the rear portions of these barrier islands as compared to the more easily erodible sandy beach and dune facies evidence by the reworking of the seaward side of these barrier islands. Beach-dune interface mapped in the GPR of these barriers show elevational offset of up to 5 m between sets of ridges. Their episodic formation is dated by volcanic ash and pumice layers deposited on these barrier islands and their associated back barrier environments (Selby and

Pullar, 1971). c) This discussion will be modified to clarify points raised above. Also an alternative LiDAR image with different rendering will be added to Figure 2. This will help to refocus this discussion on the utility of LiDAR and not the details of this specific study by emphasizing the point that how data is presented impacts identification of barrier features.

Specific Comments #2: a) P8, L8-9: this statement would be stronger if supported with examples or details of how beachface mapping can be used to infer sea level, etc. b) Nice suggestion. c) References of examples will be added to this sentence and an image of data that demonstrates beachface mapping will be added to Figure 7.

Specific Comments #3: a) P8, L14: suggest being more specific. What about the change in reflection geometry indicates storms? b) Noted that the use of "distinct geometries" is not very specific. In fact, storm-eroded beachfaces can be hard to distinguish on shape alone as evidenced in the Oliver et al. (2018b). c) A sentence will be added that states that storm eroded beachfaces display more flat-lying lower beachfaces and steeper upper beachfaces, but that the distinction form swell accreted paleo-beachface geometry is easier to detect on the basis of signal strength. Also the addition of GPR data to demonstrate this will be added to Figure 7.

Specific Comments #4: a) P8, L19-26: If the authors are going to discuss these aspects of GPR processing and interpretation (including the necessity of ground-truthing, as discussed earlier), then it is also worth noting some additional key processing steps for proper GPR interpretation. For example, migration, ideally using field common-midpoint (CMP) surveys, to determine radar velocities. b) It was very intentional not to specifically talk about processing steps, even very basic ones (as discussed in the GPR section). This is done for many reasons, one of which is that there are many different software packages, GPR brands and unit configurations (even within brands). Using the example above, some systems have transceivers and therefore cannot do common-midpoint survey; while some software packages process in terms of velocities and others use dielectric constants. We found it important to not get into specifies or

advocate for any one approach. There is already a lot of literature out about processing and even recommended steps in coastal settings. What people will use is likely a product of the equipment they have access to. In each case the user should research papers that use the same configuration and software, then use the methods from papers that provide good examples as a base to start processing their data. The end goal is to present the data in a way that best represents the subsurface stratigraphy that it is imaging and highlight the specific aspect that is the focus of discussion. To this end, we feel that gain and groundtruthing are two crucial aspects that have not been emphasized enough and therefore highlight them here. c) No addition of other processing steps will be discussed. Authors will review section explaining this decision and increase clarity about the reasoning.

Specific Comments #5: a) P10, L10: "calculations from the LiDAR data". It may be worth noting that this would not have been possible without ample stratigraphic data from sediment cores, especially given the limited GPR penetration. b) Yes to determine the total volume of barrier sands requires cores through the entire sequence. The references to calculations from the LiDAR data are specifically about the volume above MSL, similar to volume calculations for the envelope of change of modern shorelines form beach profile data. This is stated in the second mention on page 14, but oddly not the first on page 4. Thank you for drawing attention to this initial omission. c) We will add 'volume of barrier sediment supplied above mean sea level' from page 14 to the first mention of volume calculation on page 4.

Specific Comments #6: a) P12, L15: "gain control is high in the GPR data". That could just be the way in which the GPR data are shown in published form; that can be a challenge to get right. Presumably the GPR data were analyzed in high detail, and through careful tuning of gains to ensure best analysis resolution. Only those authors can speak to this. This same interpretation could be applied to criticisms of Oliver et al (2017a) noted on P14, L7-8 (LiDAR color scheme chosen for publication display). b) Gain can be a tricky step to get right and really requires cores or some idea of what is

being imaged in order to properly display the amplitude of change recorded in the GPR. Even without any groundtruthing to refer to, if the focus of the interpretation is that every beachface reflection represents a storm and ridges are formed by eolian processes, then at the very least the GPR should display the difference between beach and dune facies. We do not presume to know what the authors did with regard to the data, but can only speak to what is presented. What is known is that regardless of the detail of analysis or tuning of gains, ultimately the 'interpreted' data consists of every reflection being simply traced with no distinction of signal strength or barrier facies. With respect to the LiDAR, a similar response could be made that the presentation of the data should best reflect what the authors want the readers to focus on. While it is not known if other color schemes were trialed during analysis, the rendering chosen and topographic profiles extracted do not highlight increased accommodation space or the anomalously large sizes of the foredunes that formed during the specific time periods in question. c) Changes made to Figure 7 to clarify the point about the importance of gain. In addition to trying to demonstrate this with the core, outcrop, and GPR in Figure 3, Figure 7 will be amended to try a different approach to display the importance of gain adjustments. LiDAR with a different rendering has been added to Figure 2 to reiterate the importance of color scheme.

Technical Corrections #1: a) P2, L20-21: the point concerning collaboration between scientists with specific expertise in each of the tools (some of which [e.g., GPR or pre-processed LiDAR data] are perhaps easy to use, but not easy to use well!) described in an important one. b) This is a nice distinction that the ease of acquisition of these data, does not translate to ease of correct use (but the turn of phrase "easy to use, but not easy to use well!" is much better). c) This point will be added to the text.

Technical Corrections #2: a) Figure 5 caption: "prograded normally for a while". This is unclear, unspecific, and qualitative. (the term "normally" is applied on P11, L7 as well, and does not seem to necessarily indicate "normal [sediment surplus driven] progra-dation". Correct?). "drastic shift in evolution observed . . ." this is not clear from the

data presented, nor is it clear what would qualify as a "drastic" change b) This concept of how to term a barrier that is prograding in a consistent fashion, is something the authors discussed when it was initially referred to it as 'classic' progradation. We tried regular, uniform, ect. and agree the use of the word normal is not optimal. It is actually meant to indicate "normal [sediment surplus driven] progradation" that results in the tell-tale series of relic foredune ridges apparent in the air photograph. Within the last millennia the relic foredune ridges that formed between 1,700 and 1,000 yr BP were eroded and large transgressive dunes forms on the landward and seaward edges of the blowout. Over the last 1,000 years progradation also differs forming low-lying hummocky incipient dunes rather than larger distinct foredune ridges. In combination, this data defines a drastic shift in evolution over the last millennia as compared to previous ones. c) Changes will be made to the manuscript to clarify this point.

Technical Corrections #3: a) P11, L15: Costas et al 2016 is listed twice b) Thanks for catching this Endnote user error c) Delete one

---

## Author Response (AR1)

In following with Climate of the Past's guidelines, this response will be structured such that each point will be addressed in numerical order following the sequence: a) comments from referee, b) author's response, and c) author's changes in manuscript.

Changes addressing Chris Hein's Review:

ADRESSED General Revision #1:

- a) Consider the addition of groundtruthing as a fourth approach, equally as important as LiDAR, OSL, and GPR.
- b) While the idea of a grand slam is an appealing one, groundtruthing is not seen as a standalone technique per se. Rather coring or topographic profile collection (for example) as a means of groundtruthing remotely sensed data, is seen as an integral component of GPR and LiDAR methods. While LiDAR, GPR, and OSL can all be used individually or in various combination, it is not recommended that GPR or LiDAR is used without being groundtruthed (or at the very least state the omission and consider when interpreting the data). Coring or outcrop mapping used to ground-truth GPR and topographic profiling using levels, lasers, or GPS to ground-truth LiDAR are all techniques that can be used alone or in combination with other methods (e.g. air photograph analysis or radiocarbon dating) to study coastlines. Ultimately, our counter argument would be that groundtruthing of non-invasive subsurface data is not a critical fourth component to GOaL approach, but rather an essential element whenever remotely sensed high-resolution stratigraphic or topographic data is used (therefore embedded in GPR and LiDAR methods).
- c) No changes are planned for incorporating ground-truthing as a fourth approach. However, the introduction paragraph to GOaL methodological approach (Section 2) has been significantly rewritten to emphasize the role of ground-truthing.

ADRESSED General Revision #2:

- a) Recognize the limitations of certain field sites and conditions which may make any one of the three (or four) "hat trick" components not possible, or not the best approach for a given site.
- b) It is recognized that the combination of GPR, OSL and LiDAR is not always the best approach or even possible for a given site. However, for those sites where these techniques are able to be used this paper aims to provide insight on how to optimize their utility. We are not insinuating, nor state within the paper, that this is an "ideal" approach for all sites. What is inherent, but seems to be made more explicit, is that the optimizing of this GOaL approach was conceived for sandy prograded barriers. Following on from the discussion of the previous comment the use of GPR, OSL, and LiDAR does not preclude the use of other methods and in some cases necessitates it. Of course where there is organic material for radiocarbon dating, use that instead of or with OSL.
- c) The introductory paragraph to the GOaL methodological approach (Section 2) has been significantly rewritten to address these concerns.

ADRESSED General Revision #3:

- a) Consider adding examples from additional global sites; this does not need to be in the discussion of the three "case studies" as those are meant to be focused on single papers.
- b) This is a very good suggestion that does indeed broaden and strengthen the paper. In addition to some of references that were meant to be included, the response to this interactive discussion (both in the form of online comments and private emails) provided suggestions for other papers that will also be included. It is true that when drafting the

paper the idea was to use the three most recent examples that use GOaL specifically to study sea level, storms, and sediment supply as case studies. However, along the lines of Marc Hijma comment, it will be nice to reiterate some previous examples of successful studies mentioned earlier in the paper as reference when individually discussing study sites specifically on sea level, storms, and sediment supply.

c) Multiple additional references were included in the paper from sites spanning the globe.

**ADRESSED General Revision #4:**

- a) Consider a more measured treatment of the Oliver et al (2017a,b) studies.
- b) Since the submission of this manuscript to Climate of the Past, an extensive comment on Oliver et al. (2017a) has been published and a similarly detailed discussion paper on Oliver et al. (2017b) has been accepted with minor modifications (Dougherty, 2018a,b). These comment papers allow a nuanced conversation of the data and interpretations presented in each that was simply not possible nor appropriate for this technical note. This allows us to refer to these comment papers for a fuller discussion when identifying the potential pitfalls encountered when GPR, OSL and LiDAR are not used optimally. This technical not will focus specifically on how interpretations can be questioned when GPR data is not groundtruthed with cores as well as when rendering of LiDAR (and topographic profiles extracted from the remotely sensed data) masks or distracts from important aspects of the morphology. The importance of groundtruthing is agreed, as per the discussion above, and presnetaion publicly does matter if it is influencing interpretation (see discussion in Specific Comment #6 below). While it is uncomfortable to critique specific studies, it is necessary in order to have a rigorous scientific debate. Because of the critical nature of this aspect of the paper, it was important to us that it was published in a way that the authors of the Oliver et al. (2017a, b) papers could comment or correct anything that might be misleading. We are grateful for Climate of the Past's open access and interactive review process which allowed this option. Authors associated with both papers were made aware of this technical note addressing methods used in their papers before it was submitted and it is known that the lead author has viewed this Climate of the Past submission (as informed by ResearchGate).

**3.2 Storms**

To determine a storm record requires eroded paleo-beachfaces to be clearly identified within the stratigraphy. This requires coring of these storm layers and using these cores to groundtruth the GPR and make gain adjustments. Failure to do this can result in ambiguous interpretation of storm plaeo-beachfaces. A new example of GPR data will be added that shows GPR data with high gain applied masking the storm layers. This same data with the gain reduced according to an overlain core reveals obvious storm layers. This will demonstrate the potential to interpret an over exaggerated storm history for sites that do not core or adjust gain, such as Oliver et al. (2017b).

**3.3 Sediment supply and coastal evolution**

In order to discuss changes in evolution through time and not confuse these with alongshore variations, a single shore-perpendicular transect line from the oldest to the youngest part of the barrier is optimal. This GOaL paper advocates for using LiDAR to determine where the best location is to collect this transect. Then collect GPR along this transect and use this morphostratigraphy to target the best location to collect OSL. We agree with the reviewer that integrating two or more parallel lines improves the robustness as well as allowing a discussion about alongshore variation through time. However, this is not our point or the reason for the multiple lines drawn in Figure 7d. Rather these lines were extracted from the rerendered LiDAR to demonstrate how it can be used to determine the best location to extract the best transect to span the entire history. This process clearly shows that the

western profile is the most complete. Comparing this one transect to those presented by Oliver et al. (2017a) identifies gaps within the morphologic data at points where the evolution shifts towards the beginning of barrier inception and in the most recent period. The LiDAR shows the accommodation space changes through time and identifies that anomalously large foredune ridges formed during this time. However, GPR and OSL were not collected for these areas of interest. The gaps in the topographic record, GPR stratigraphy and OSL chronology during these two points in time raises the question of whether the interpretation of halted progradation is due to ceased sediment supply or just a result of a lack of data. At the very least identifying that these gaps exist and considering the implication is crucial to any discussion about barrier evolution and the role of sediment supply. This does not negate sizable amount of data presented for the other area of the work that no doubt went into collecting it. Rather that following the simple order and suggestions presented in this GOaL methodology might have helped optimize this dataset by first targeting shifts in evolution using LiDAR morphology, detail how these sections of the barrier formed using detailed stratigraphy from GPR and then dating the timing of these changes using targeted OSL samples.

c) The two sections referring to the Oliver et al. (2017a,b) studies were redrafted so that specifics to the study sites are minimized and transferrable lessons for the best practice use of the GOaL techniques are emphasized (as suggested by the reviewer).

**ADRESSED Specific Comments #1:**

- a) P6, L9 & Figure 2: this is focused on the utility of LiDAR, not the details of this study. However, it is not clear that the multiple sets of "prograded barrier islands" shown here were never a single island / beach-ridge plain. This is a great example of possible reworking of a non-continuous record, a limitation in reconstructing the evolutionary history of a progradational site, or the paleoenvironmental records contained within. This in fact may be a case for the use of subsurface data (GPR) to search for, e.g., landward-dipping beds at the landward side of each of those "islands" to try to infer if they formed as separate transgressive-regressive islands. LiDAR data here may in fact tell an incomplete story.
- b) The main point of this section was to demonstrate how LiDAR can provide improved images of the morphology of coastal system and their surroundings to inform where to collect GPR. This approach was used at Rangitaiki Plains, but the GPR aspect was excluded so that LiDAR could be the focus. Based on this comment, it seems pertinent to add that this informed GPR collection. The combination of these data, along with previous studies of the area, was used to determine the evolution of this complex coastal system. The evidence for this series of prograded barrier islands is that the landward extent of each naturally transition into backbarrier deposits. These cohesive muds were more resistant to erosion and therefore preserved the morphology of the rear portions of these barrier islands as compared to the more easily erodible sandy beach and dune facies evidence by the reworking of the seaward side of these barrier islands. Beach-dune interface mapped in the GPR of these barriers show elevational offset of up to 5 m between sets of ridges. Their episodic formation is dated by volcanic ash and pumice layers deposited on these barrier islands and their associated back barrier environments (Selby and Pullar, 1971).
- c) This discussion has been modified to clarify points raised above. Also an alternative LiDAR image with different rendering was added to Figure 2 in order to help to refocus this discussion on the utility of LiDAR to guide GPR collection and the importance of rendering/presentation on the identification of barrier features.

ADRESSED Specific Comments #2:

- a) P8, L8-9: this statement would be stronger if supported with examples or details of how beachface mapping can be used to infer sea level, etc.
- b) Nice suggestion.
- c) References of examples were added to this sentence and an image of data that demonstrates beachface mapping was added to Figure 7(d,e,).

ADRESSED Specific Comments #3:

- a) P8, L14: suggest being more specific. What about the change in reflection geometry indicates storms?
- b) Noted that the use of "distinct geometries" is not very specific. In fact, storm-eroded beachfaces can be hard to distinguish on shape alone as evidenced in the Oliver et al. (2018b).
- c) A sentence was added that states that storm steeper upper beachfaces with a strong reflection that makes it distinct form swell accreted paleo-beachface geometry. GPR data was also added to Figure 7 to demonstrate this.

**ADRESSED Specific Comments #4:**

- a) P8, L19-26: If the authors are going to discuss these aspects of GPR processing and interpretation (including the necessity of ground-truthing, as discussed earlier), then it is also worth noting some additional key processing steps for proper GPR interpretation. For example, migration, ideally using field common-midpoint (CMP) surveys, to determine radar velocities.
- b) It was very intentional not to specifically talk about processing steps, even very basic ones (as discussed in the GPR section). This is done for many reasons, one of which is that there are many different software packages, GPR brands and unit configurations (even within brands). Using the example above, some systems have transceivers and therefore cannot do common-midpoint survey; while some software packages process in terms of velocities and others use dielectric constants. We found it important to not get into specifies or advocate for any one approach. There is already a lot of literature out about processing and even recommended steps in coastal settings. What people will use is likely a product of the equipment they have access to. In each case the user should research papers that use the same configuration and software, then use the methods from papers that provide good examples as a base to start processing their data. The end goal is to present the data in a way that best represents the subsurface stratigraphy that it is imaging and highlight the specific aspect that is the focus of discussion. To this end, we feel that gain and groundtruthing are two crucial aspects that have not been emphasized enough and therefore highlight them here.
- c) A paragraph has been added to the GPR section to explain as well as a couple of sentences to the paragraph about gain.

ADRESSED Specific Comments #5:

- a) P10, L10: "calculations from the LiDAR data". It may be worth noting that this would not have been possible without ample stratigraphic data from sediment cores, especially given the limited GPR penetration.
- b) Yes to determine the total volume of barrier sands requires cores through the entire sequence. The references to calculations from the LiDAR data are specifically about the volume above MSL, similar to volume calculations for the envelope of change of modern shorelines form beach profile data. This is stated in the second mention on page 14, but oddly not the first on page 4. Thank you for drawing attention to this initial omission.
- c) The clarification of 'volume of barrier sediment supplied above mean sea level' was added to the first mention of volume calculation on page 4.

**ADRESSED Specific Comments #6:**

- a) P12, L15: "gain control is high in the GPR data". That could just be the way in which the GPR data are shown in published form; that can be a challenge to get right. Presumably the GPR data were analyzed in high detail, and through careful tuning of gains to ensure best analysis resolution. Only those authors can speak to this. This same interpretation could be applied to criticisms of Oliver et al (2017a) noted on P14, L7-8 (LiDAR color scheme chosen for publication display).
- b) Gain can be a tricky step to get right and really requires cores or some idea of what is being imaged in order to properly display the amplitude of change recorded in the GPR. Even without any groundtruthing to refer to, if the focus of the interpretation is that every beachface reflection represents a storm and ridges are formed by eolian processes, then at the very least the GPR should display the difference between beach and dune facies. We do not presume to know what the authors did with regard to the data, but can only speak to what is presented. What is known is that regardless of the detail of analysis or tuning of gains, ultimately the 'interpreted' data consists of every reflection being simply traced with no distinction of signal strength or barrier facies. With respect to the LiDAR, a similar response could be made that the presentation of the data should best reflect what the authors want the readers to focus on. While it is not known if other color schemes were trialed during analysis, the rendering chosen and topographic profiles extracted do not highlight increased accommodation space or the anomalously large sizes of the foredunes that formed during the specific time periods in question.
- c) Changes made to Figure 7 to clarify the point about the importance of gain. In addition to trying to demonstrate this with the core, outcrop, and GPR in Figure 3, Figure 7 was amended to try a different approach to display the importance of gain adjustments. LiDAR with a different rendering has been added to Figure 2 to reiterate the importance of color scheme.

ADRESSED Technical Corrections #1:

- a) P2, L20-21: the point concerning collaboration between scientists with specific expertise in each of the tools (some of which [e.g., GPR or pre-processed LiDAR data] are perhaps easy to use, but not easy to use well!) described in an important one.
- b) This is a nice distinction that the ease of acquisition of these data, does not translate to ease of correct use (but the turn of phrase "easy to use, but not easy to use well!" is much better).
- c) This point was added to the text and Hein personal communication 19 March 2018 was cited.

**ADRESSED Technical Corrections #2:**

- a) Figure 5 caption: "prograded normally for a while". This is unclear, unspecific, and qualitative. (the term "normally" is applied on P11, L7 as well, and does not seem to necessarily indicate "normal [sediment surplus driven] progradation". Correct?). "drastic shift in evolution observed . . ." this is not clear from the data presented, nor is it clear what would qualify as a "drastic" change
- b) This concept of how to term a barrier that is prograding in a consistent fashion, is something the authors discussed when it was initially referred to it as 'classic' progradation. We tried regular, uniform, ect. and agree the use of the word normal is not optimal. It is actually meant to indicate "normal [sediment surplus driven] progradation" that results in the tell-tale series of relic foredune ridges apparent in the air photograph. Within the last millennia the relic foredune ridges that formed between 1,700 and 1,000 yr BP were eroded and large transgressive dunes forms on the landward and seaward edges of the blowout. Over the last

1,000 years progradation also differs forming low-lying hummocky incipient dunes rather than larger distinct foredune ridges. In combination, this data defines a drastic shift in evolution over the last millennia as compared to previous ones.

c) The word 'normally' was change to 'uniformly' throughout.

ADRESSED Technical Corrections #3:

- a) P11, L15: Costas et al 2016 is listed twice
- b) Thanks for catching this Endnote user error
- c) Deleted one

Changes addressing Zhixiong Shen's Review:

ADRESSED General Comment #1:

a) One advantage of combining Li-DAR topography and GPR not mentioned yet is that the former is very useful for elevation correction of the latter. However, some common pitfalls of the individual technique are not mentioned, which makes the strategy practically less useful to follow.

b) LiDAR is useful for rough topographic correction. Due to the fact that I use most of my data to try and infer sea level from heights of beachfaces and storms form their geometry, I like to use the most accurate topographic correction possible. To eliminate the uncertainties associated with LiDAR and extracting the exact transect line form the image that the GPR was collected along (usually the two data sets are acquired on different dates allowing possible discrepancies), I prefer the old school method of levelling or laser levelling in the actual elevations of the GPR transect line. This should be done at the time of collection and include a survey of the active beach and tie in any existing benchmarks. This is standard along with coring to ground-truth depth to water table on the day of GPR collection.

c) A sentence was added to the GPR section that states the utility of LiDAR to roughly topo correct GPR data, but that a proper survey should be conducted alongside GPR collection.

ADRESSED General Comment #2:

a) GPR data collection and interpretation depend not only on gain, but also on the frequency of radar, antenna shielding, spacing of traces, and speed of radar in sediments of different nature. I am not sure why the note specifically picked gain, but not others in the recommendation.

b) True there are many different set up and settings when it comes to collecting GPR. What is ultimately chosen depends greatly on the access to gear, the field conditions, the scientific question to be answered, etc. At this point there are plenty of papers and books about GPR that people can turn to in order to understand how GPR works and what settings are best for each particular study. The aim of this paper was to show the potential of optimizing the use of GPR in combination with OSL and LiDAR, not a' how-to' guide with specifics for acquiring and analysing each data set. Rather this paper works on the basis that readers have a standard knowledge of, and some experience with, these techniques. As advocated in the paper, it is important to lean from or collaborate with experts using each of these three methods as they are specific fields within geophysics, geochronology and remote sensing. c) A paragraph has been added to the GPR section to explain as well as a couple of sentences to the paragraph about gain.

b) With respect to GPR, the units have become so affordable and user-friendly it has become relatively easy to access a machine, turn it on, and get data. While this is absolutely fantastic for the expansion of its use in many fields, there is a danger that without proper knowledge and training data can be collected incorrectly and/or misinterpreted. For the most part in the field of coastal research this is not the case, probably because it has been used in this field for so long with many people and papers to turn to. Therefore, most papers have the basics correct. The reason gain is singled out is for two reasons: 1) applying a high gain (as well as highlighting every reflection) is one very common occurrence in coastal data and 2) it is an incredibly important adjustment when trying to extract storm and sea level records from sandy barriers. I have increasingly noticed that along with the proliferation of GPR use, more papers and conference presentations don't have the gain adjusted so that the signal strength aptly reflects the contrast in the stratigraphy. I have seen high gain inhibit the identification and extraction of a sea level curve from otherwise good data and result in an exaggeration of the storm frequency and impact as demonstrated in Oliver et al. (2017b).

c) A paragraph has been added to the GPR section to explain as well as a couple of sentences to the paragraph about gain.

ADRESSED General Comment #3:

a) OSL age determination is affected by many assumptions about bleaching, distribution of radioactive sources in the sediment, water content variation, post depositional disturbance, disequilibrium in the uranium and thorium decay series, and cosmic radiation (often a very important component to the total radiation a beach sample received) change because of change of overlying sediment thickness. The choice of appropriate age model does not handle all these complications.

b) These are important considerations and will be assessed for incorporation within the manuscript.

c) A paragraph has been added and scope of the section shifted.

ADRESSED General Comment #4:

a) One more recommendation about OSL date is that the ages should be reported in a way to enable comparison across different publications. This is because OSL ages refer to the time before OSL measurement and the measurement time must be reported to ensure comparison. As an example of inappropriate reporting OSL data, I noted that the note used 'BP' as a unit for OSL data, which suggests to me that these OSL data refers to AD 1950 following the most common use of BP in the geochronology community. However, my sense is that I am reading the unit 'BP' in the note wrongly.

b) Another good point, this will be addressed in the final draft of the manuscript if accepted.

c) This has been included in the new paragraph.

ADRESSED Specific comments: P4, L4: add 'can' before 'be utilized' -reworded for clarity P4, L10: add 'of' before 'coastal' toward the end of the line

-recommended change made

P6, L4: parenthesis for reference not correctly used

-good eye, changed

P6, L12-13: delete 'in the' between the two lines

-again thank you for catching this, changed

P9, L11: replace 'bleaches' by 'can bleach', and 'any' by 'light-sensitive'

- nice clarification, recommended change made

P9, L15: replace 'accumulation period' by 'burial period'

-better, recommended change made

P11, L5-6: 'In order to decipher the timing of this shift, the aerial imagery was used to target the changes in morphology and GPR to locate corresponding differences in the underlying stratigraphy'. Can this be shown in Fig. 5b?

-The figure caption has been rewritten to describe how it is shown in Fig. 5b.

P11, L14-15: repeating reference

-one deleted

P11, L20-21: 'Initially this complicated spit system did not appear as an ideal site to extract a sea level history'. What is the reason for this?

-The idea is that it looks more like longshore spit progradation, but the LiDAR displays a section of the barrier that shows foredune ridges identifying seaward progradation. However, after reading this comment, I think that this is not necessary and therefore this line has been deleted.

P14, L7: sentence toward the end does not seem to be complete

-completed

P16, L4: replace 'form' by 'from'

-thanks for catching this dyslexic tendency, changed

P16, L5: replace 'intern' by 'in turn'

-thanks, changed

Fig 1: the OSL data seem represented by circles filled by brown, but not open black circles as indicated in the caption.

-changed wording to clarify

Fig 2: legend in 2b is not legible

-figure has been modified to show that this legend is the same as the larger one legible in 2d

Fig 6: the thin dashed lines in 6b are not interpreted.

-the figure caption has been changed to include definition of these

Why are there two different y-scales in 6c?

-left is the elevation of the markers and the right is the elevation of the overlying dunes

What is the difference between berm markers and beach-dune markers? Are the latter beach/dune boundary?

-These are both in relation to the upper beach that was mapped and used as potential indicators of sea-level position within the barrier stratigraphy (i.e. beach backshore and upper foreshore).

[revised manuscript text omitted]

---

## Referee Report (RR1)

**Manuscript Review**

**Title:** Technical note: Optimizing the utility of combined GPR, OSL, and LiDAR (GOaL) to extract paleoenvironmental records and decipher shoreline evolution
**Authors:** Dougherty, A.J., Choi, J.-H., Turney, C.S.M., and Dosseto, A.

**General Comments**

This is a re-review of a manuscript reviewed originally in March 2018. Please note that all page and line numbers given here are in reference to the version of the manuscript presented with changes tracked.

The authors have done a very nice job of responding to comments and concerns, and updating the manuscript in accordance with public reviews and comments. In particular, the authors' discussion around my first general edit ("consider the addition of ground-truthing as a fourth approach") was well thought-out, and the edits to the manuscript appropriately address this. I thank the authors for their thoughtful consideration of reviewer critiques and hope that my input was of use as they improved this manuscript.

My only response comment is to the final sentence of the Author Response to General Revision #2. The authors make a strong case here for the utility of OSL age dating (and a sub-set of the authors contributed to what I find to be a very compelling paper demonstrating the potential advantage of OSL dating over radiocarbon dating in Oliver et al., 2015 [Holocene]); however, I would be very hesitant to discount the use of radiocarbon dating of organic matter in all cases, even when a setting lends itself to determination of deposition age estimates using OSL. Radiocarbon remains a scientific standard and can be accomplished on very small (<20 ug) samples, providing the provenance of the organic matter is well constrained. Commercially, 3-4 radiocarbon samples can be analyzed for the price of one OSL sample, and in a matter of weeks rather than 6-12 months. Each approach has its place, and the authors may consider recognizing this better in the manuscript (this is done somewhat on P5, L21-23).

I find the manuscript much improved and, though we may still quibble on some points, certainly suitable for publication. The authors have done a particularly good job in:

1) clarifying the statement of purpose of the manuscript (P5, L2-6).

2) describing their treatment of each of the methods and clarifying the reasoning behind presenting a (relatively) low level of detail on each the technicalities of each underlying approach and on the methods for applying and analyzing associated data.

3) presenting clear best practices for use of each of the GoAL methods independently and in conjunction for these specific types of settings and studies

4) presenting a much improved Fig 7 with a clearer purpose and very nice demonstration, using their own data, of how proper use of gain (and I note again, with

ground-truthing) can improve analysis of GPR data (as opposed to simply pointing out perceived faults in the Oliver et al 2017 [Geomorphology] paper).

5) presenting a much more measured treatment of the Oliver et al 2017 [Geomorphology] and Oliver et al 2017 [Marine Geology] manuscripts. In particular, the authors benefitted from the publication of several Discussion articles in the months during which this manuscript was in review and revision, which allowed the authors to focus their discussion of these two prior studies on lessons for best practices in data collection and utilization. This was one of my primary critiques in the original manuscript and the authors have handled it very nicely.

My remaining suggestions are all very minor:

1) The manuscript would benefit from some additional detailed copy editing to correct issues such as (but not limited to):
(a) use of data as singular (perhaps the plurality of the word "data" is a personal preference rather than grammatical rule?) (e.g., P2, L30);
(b) missing closed parentheses (P3, L7)
(c) hyphenation of "three-step" (P4, L11)change "no" to "not" (P5, L22)
(d) comma missing after "used" (P5, L33)
(e) change "coasts" to "coastal" (P13, L21)
(f) consider not capitalizing "penetrating" and "radar" (P15, L7)
(g) "rendering of the LiDAR as well" (P26, L8) and "no GPR was" (P26, L27): consider adding "data were" after both "LiDAR" and "GPR"
(h) add comma after "how" (P26, L33)
(i) incomplete sentence in P27, L18-19.

2) Several suggested wording changes for clarity:
(a) change "coastal plains" (a geomorphic/geographic feature) to "coastal settings" (P5, L28)
(b) "there has been relatively little discussion about gain" (P15, L33) – where has there been little discussion? In the literature? Needs clarity.
(c) sentence structure in P17, L15-17 is very confusing
(d) "deep" coring (P18, L1): needs to be defined. Is this word even necessary?
(e) not sure "decipher" is the best word choice (P22, L18); consider "differentiate"
(f) consider adding something along the lines of ", therefore leading to incorrect or incomplete interpretations" after "overlooked" (P27, L1)
(g) Use of "evolution" is somewhat unclear (P27, L11).

3) P6, L123-14: the authors may consider a few sentences about other, newer methods/technologies for mapping morphology. I am specifically thinking about drone-based "structure from motion" (sfm). Unlike the earlier author-reviewer discussion we shared about the need for ground-truthing, this is not a significant consideration. However, it may be worth the authors mentioning the existence of other techniques

which could produce lidar-like and lidar-quality data across similarly large areas. Lidar data are still not available for large parts of the world. While, as the authors note, lidar benefits from penetrating vegetation which may obscure morphology (P6, L23-24), there are many coastal settings (e.g., coarse-grained and/or high-latitude beach ridges where vegetation is minimal or non-existent) where progradational beach- and foredune- ridge plains could be easily (at less expense and simplified logistics) mapped using, e.g., sfm. The point is that lidar is one tool of several for high-resolution, large-spatial extent topographic mapping, and data produced by another method may be equally valid for use in a GOaL-like approach.

4) P21, L24-25 and P27, L21-22: I would be remiss if I did not highlight the work in a very challenging environment to produce a sea-level curve using OSL dating and GPR, as well as detailed topographic mapping (using RTK-GPS as opposed to lidar) by my colleague Dr. Billy (Billy et al., 2015, Geomorphology). This paper is already cited in this manuscript, so this is not fishing for a citation, but rather highlighting a recent study that successfully used the exact approach proposed by the authors.

5) P17, L22-26: I recognize this addition is in response to another reviewer. I agree with this and have struggled with it myself, especially when presenting both radiocarbon (in years B.P.) and OSL dates in the same discussion. The difference matters little when discussing dates of >10,000 years, but can matter a lot for the last 1000 years. For those shorter time periods, simply converting the radiocarbon ages and presenting all dates in years CE is one solution my reviewers have agreed on.